# A genome-wide CRISPR-Cas9 knockout screen identifies FSP1 as the warfarin-resistant vitamin K reductase

Da-Yun Jin [1], Xuejie Chen [1], Yizhou Liu [2], Craig M. Williams [2], Lars C. Pedersen [3], Darrel W. Stafford [1] & Jian-Ke Tie [1]✉

Vitamin K is a vital micronutrient implicated in a variety of human diseases. Warfarin, a vitamin K antagonist, is the most commonly prescribed oral anticoagulant. Patients overdosed on warfarin can be rescued by administering high doses of vitamin K because of the existence of a warfarin-resistant vitamin K reductase. Despite the functional discovery of vitamin K reductase over eight decades ago, its identity remained elusive. Here, we report the identification of warfarin-resistant vitamin K reductase using a genome-wide CRISPR-Cas9 knockout screen with a vitamin K-dependent apoptotic reporter cell line. We find that ferroptosis suppressor protein 1 (FSP1), a ubiquinone oxidoreductase, is the enzyme responsible for vitamin K reduction in a warfarin-resistant manner, consistent with a recent discovery by Mishima et al. FSP1 inhibitor that inhibited ubiquinone reduction and thus triggered cancer cell ferroptosis, displays strong inhibition of vitamin K-dependent carboxylation. Intriguingly, dihydroorotate dehydrogenase, another ubiquinone-associated ferroptosis suppressor protein parallel to the function of FSP1, does not support vitamin K-dependent carboxylation. These findings provide new insights into selectively controlling the physiological and pathological processes involving electron transfers mediated by vitamin K and ubiquinone.

Vitamin K was identified as a fat-soluble factor that was required for preventing hemorrhages in chickens[1]. The linkage between vitamin K and coagulation was further strengthened by the discovery of vitamin K-dependent (VKD) carboxylation of glutamic acid residues in prothrombin and other procoagulant and anticoagulant proteins[2]. As new VKD proteins were discovered, the importance of VKD carboxylation to a variety of non-coagulation functions was revealed. Disorders of vitamin K metabolism are now associated with various human diseases[3–6].

VKD carboxylation is an essential post-translational modification catalyzed by gamma-glutamyl carboxylase (GGCX). Carboxylation is required for the biological functions of proteins that control blood coagulation, vascular calcification, bone metabolism, and other important physiological and pathological processes. Concomitant with carboxylation, reduced vitamin K (vitamin K hydroquinone, $KH_2$) is oxidized to vitamin K epoxide ($K_{epo}$), which must be converted back to $KH_2$ through a two-step reduction by the enzymes vitamin K epoxide reductase (VKOR) and vitamin K reductase (VKR) in a pathway known as the vitamin K cycle. Although the enzymatic activity of the vitamin K cycle was first discovered in the 1970s[7], identification of these enzymes required more than half a century.

GGCX was cloned and purified by our laboratory in 1991[8,9], and the gene encoding VKOR was identified independently by our laboratory and that of Oldenburg in 2004[10,11]. Identifying GGCX and VKOR made it possible to link genetic variations of these genes with their corresponding clinical phenotypes and thus provided invaluable

[1]Biology Department, University of North Carolina at Chapel Hill, Chapel Hill, NC 27599, USA. [2]School of Chemistry and Molecular Biosciences, University of Queensland, Brisbane, QLD 4072, Australia. [3]Genome Integrity and Structural Biology Laboratory, National Institute of Environmental Health Sciences, National Institutes of Health, Research Triangle Park, NC 27709, USA. ✉e-mail: jktie@email.unc.edu

understanding into the control of bleeding and nonbleeding disorders[12]. However, despite significant efforts in the past several decades, the identity of VKR remained elusive.

In this study, we developed a unique universal VKD apoptotic reporter cell line that is susceptible to VKD carboxylation. Combined with CRISPR-Cas9 genome-wide knockout screening, we identify ferroptosis suppressor protein 1 (FSP1) (also known as AIFM2; Apoptosis Inducing Factor Mitochondria-associated 2) to be responsible for warfarin-resistant vitamin K reduction. During the preparation of our manuscript, we became aware of the article published by Mishima et al., who screened a number of naturally available vitamin compounds for preventing ferroptosis to reach the same conclusion[13]. Nevertheless, our VKD apoptotic reporter cell line, in combination with the emergence of genome-wide CRISPR-Cas9 knockout library screening, now makes it possible to identify other unknown enzymes associated with the vitamin K cycle.

## Results

### VKD apoptotic reporter cell line for large-scale screening

VKD carboxylation is a nonlethal post-translational modification with no apparent phenotypic consequences in cell growth, making it impossible for genome-wide screening to assist in identification of unknown enzymes in the vitamin K cycle. Therefore, developing a VKD apoptotic reporter cell line for large-scale screening of enzymes associated with VKD carboxylation was critical to underpinning the genome-wide screening approach. The reporter cell line described herein was inspired by the agonistic antibody-mediated Fas receptor associated extrinsic pathway of apoptosis[14,15]. The interaction between Fas and an agonistic anti-Fas antibody results in the recruitment of several signaling proteins, such as Fas-associated death domain protein (FADD) and procaspase-8, to form a complex designated death-inducing signaling complex, and activate the Caspase 8-mediated apoptosis pathway (Fig. 1a). To link the non-phenotypic VKD carboxylation with Fas-mediated apoptosis, we fused a VKD domain FIXgla (the Gla domain of factor IX) to the extracellular N-terminus of Fas (Fig. 1b and Supplementary Fig. 1a), and stably expressed this reporter protein in HEK293 cells. We anticipated that the monoclonal antibody that specifically recognizes carboxylated FIXgla, of the chimeric reporter protein FIXgla-Fas, would trigger the Fas-associated apoptosis pathway, similarly to that of the Fas agonistic antibody.

Our results show that introduction of FIXgla to the N-terminus of Fas does not affect its expression and plasma membrane localization in HEK293 cells (Fig. 1c). Importantly, the chimeric reporter protein FIXgla-Fas can be carboxylated when cells were fed vitamin K, and conversely carboxylation was abolished by warfarin inhibition. Strikingly, it was observed that the FIXgla-Fas/HEK293 reporter cells must be maintained in the warfarin-containing medium, as the residual vitamin K in the cell culture medium was suitably sufficient to stimulate apoptosis of the reporter cells, even without antibody incubation. It was also noticed that VKD apoptosis of the FIXgla-Fas/HEK293 reporter cell was highly sensitive and dose-dependent on vitamin K (LC$_{50}$, 6.5 nM) (Fig. 1d), and the carboxylation of FIXgla-Fas occurred in a time-dependent manner (Fig. 1e). Upon addition of vitamin K, most cells detached from the culture flask within 24 h (Fig. 1f), thus providing a robust positive selection mechanism for identifying vitamin K-associated genes in the genome-scale screening process. Consistent with Fas-agonistic antibody activated apoptosis, VKD apoptosis of the FIXgla-Fas/HEK293 cells were also mediated by the Caspase-dependent pathway (Fig. 1g). This was further confirmed by mutating residue D260, an important residue for Fas-FADD interaction[16], or by deleting the death domain (DD) of Fas, which abolished VKD apoptosis (Supplementary Fig. 1b). Together these results suggested that the FIXgla-Fas/HEK293 reporter cell line was susceptible to VKD carboxylation, and that disruption of the enzymes associated with the vitamin K cycle would prevent apoptosis i.e., allow functional

screening for any unknown enzymes associated with the vitamin K cycle via a positive selection mechanism[17].

### A genome-wide screen reveals FSP1 as the warfarin-resistant VKR

To identify VKR, we used a genome-wide CRISPR-Cas9 knockout library (Brunello), which is composed of 76,441 single guide RNAs (sgRNAs) targeting 19,114 genes in the human genome[18]. The Brunello lentiviral library was transduced into the FIXgla-Fas/HEK293 reporter cells, stably expressing Cas9, at a low multiplicity of infection (MOI, 0.3) to ensure that most cells received only one stable integrated sgRNA for genetic perturbation (with a coverage of >500 cells expressing each sgRNA)[17] (Fig. 2a). Following puromycin selection for 7 days, $5 \times 10^7$ cells were harvested as a control before vitamin K selection. To the remaining cells, we included 11 μM vitamin K and 5 μM warfarin in the cell culture medium for functional screening of the warfarin-resistant VKR. Warfarin was used to eliminate any possible contributions from warfarin-sensitive VKR to the functional screening. As warfarin also inactivates VKOR for vitamin K recycling[19], we used a higher concentration of vitamin K (~10-fold higher than the maximum concentration for VKD apoptosis, Fig. 1d) for functional screening. Cells expressing sgRNAs targeting genes in the vitamin K cycle would abolish VKD carboxylation of the reporter protein FIXgla-Fas, thus resulting in the survival of the cells (Fig. 2b), whereas those expressing sgRNAs targeting genes irrelevant to VKD carboxylation were expected to be depleted by VKD apoptosis. Surviving cells were then harvested and subjected to genomic DNA analysis by next-generation sequencing. The most significant hits from two independent screens were examined to account for stochastic noise. Top-hit candidates were assessed based on fold enrichment of sgRNA reads and the number of unique sgRNAs enriched per gene (Fig. 2c).

Consistent with our initial design of the VKD apoptotic reporter cell line (Fig. 1), sgRNAs targeting proteins in the Fas-associated apoptosis pathway (8 of the 10 proteins in the STRING Protein-Protein Interaction Networks) (Supplementary Fig. 2) were significantly enriched in the vitamin K-treated sample (Supplementary Data 1). Most of these genes were targeted by more than one sgRNA and essential genes such as Fas, FADD, Caspase-3, and Caspase-8 were targeted by all four sgRNAs in the library, which further supported the hypothesis that VKD carboxylation of FIXgla-Fas triggered Fas-associated apoptosis. Additionally, sgRNAs targeting several calcium-dependent proteins were enriched in the vitamin K-treated sample, supporting the role of calcium ions in Fas-mediated apoptotic pathway[20]. Using fold enrichment of sgRNAs targeting GGCX as a threshold, we selected candidate hits of warfarin-resistant VKR based on their potential protein functions from the UniProtKB database (Supplementary Data 2). To validate the candidate genes and minimize possible off-target effects, the top two enriched sgRNAs of each gene were used to knockout the candidate genes in an alternate VKD carboxylation reporter cell line (FIXgla-Met.Luc/HEK293). Lentivirus containing Cas9 and the corresponding sgRNA were transduced into FIXgla-Met.Luc/HEK293 cells, along with the positive control (sgRNA targeting GGCX), and the negative control (non-targeting sgRNA) for gene knockout. Our results revealed that both sgRNAs that target FSP1 significantly decreased VKD carboxylation when the reporter cells were incubated with vitamin K in the presence of warfarin (Fig. 2d). Moreover, the sgRNA of FSP1-2 knocked out carboxylation activity to a similar level as knocking out GGCX. This result suggested that FSP1 was responsible for the warfarin-resistant VKR activity, which is consistent with a recent study[13].

We then deployed the sgRNA of FSP1-2 to knock out the endogenous *fsp1* gene to obtain a stable gene-knockout cell clone. When *fsp1* was knocked out from HEK293 cells (exon 4 was targeted as predicted, Supplementary Fig. 3), no FSP1 protein was detected (Fig. 2e), and VKR activity was significantly reduced (Fig. 2f). Furthermore,

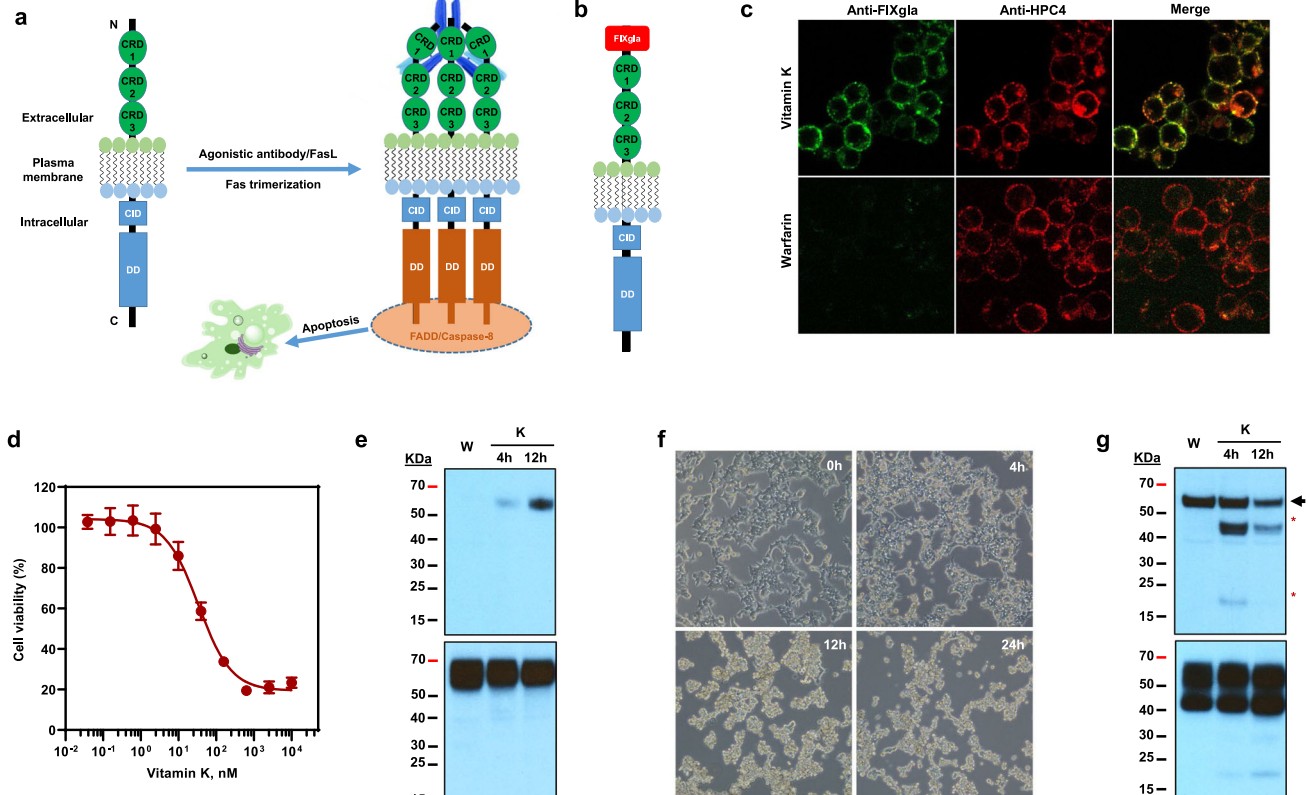

**Fig. 1 | Characterization of the vitamin K-dependent apoptotic reporter cell line. a** Schematic diagram of activation of Fas-mediated apoptosis. Fas ligand (FasL) or Fas agonistic antibody activates Fas by forming a trimer and death-inducing signaling complex triggering the extrinsic pathway of apoptosis. CRD cysteine-rich domain, CID calcium-inducing domain, DD death domain, FADD Fas-associated protein with death domain. Fas agonistic antibody was indicated as Y shape in blue on the back of the CRD domains of activated Fas. **b** Diagram of the chimeric reporter protein FIXgla-Fas with the Gla domain of factor IX fused to the extracellular N-terminus of Fas. **c** Immunofluorescence confocal microscope imaging of HEK293 cells stably expressing the reporter protein FIXgla-Fas. FIXgla-Fas/HEK293 reporter cells were incubated with 11 μM vitamin K (top) or 5 μM warfarin (bottom) for 12 h. Cells were co-immunostained with FITC-labeled anti-carboxylated FIXgla antibody (green) and APC-labeled anti-HPC4 antibody (red). **d** VKD apoptosis of FIXgla-Fas/HEK293 reporter cells. The reporter cells were incubated with increasing concentrations of vitamin K for 24 h and the cell viability was determined using cell-

counting Kit-8. Data are presented as mean ± SD of five independent experiments (*n* = 5). **e** Immunoblotting of the carboxylated (top, probed by anti-FIXgla antibody) and total (bottom, probed by anti-HPC4 antibody) reporter protein FIXgla-Fas after the reporter cells were incubated with 11 μM vitamin K at different time points as indicated. Reporter cells incubated with 5 μM warfarin (W) was used as a control. **f** Inverted microscope imaging of FIXgla-Fas/HEK293 reporter cells at different time points after treated with 11 μM vitamin K. **g** Immunoblotting of caspase-dependent apoptosis of FIXgla-Fas reporter cells. Reporter cells were incubated with 5 μM warfarin (W) or 11 μM vitamin K (K) at the indicated time points. Top: Activation of caspase-8 was probed by anti-caspase-8 antibody. Full-length caspase-8 was indicated by an arrowhead, activated caspase-8 was indicated by asterisks. Bottom: total FIXgla-Fas reporter protein expression probed by anti-Fas antibody. Similar results were observed at least three times as shown in Fig. 1f and twice as shown in Fig. 1e, g.

introducing the *fsp1* gene back into the knockout cells resulted in restoration of VKR activity. The vitamin K reduction activity of FSP1 was further confirmed by directly following the production of $KH_2$ from vitamin K using the conventional HPLC-based in vitro activity assay. Heterologous expression of FSP1 in HEK293 cells robustly increased $KH_2$ production (Fig. 2g). Additionally, the in vitro VKR activity of FSP1 was resistant to warfarin inhibition (Fig. 2h). Together, these results support the concept that FSP1 is the warfarin-resistant VKR that is essential for VKD carboxylation.

## The multiple pathways for vitamin K reduction

Our in vitro studies show that VKOR also can reduce vitamin K to $KH_2$[21,22]. To further clarify the contribution of VKOR and FSP1 to vitamin K reduction in a cellular milieu, we examined the carboxylation efficiency of FIXgla-PC/HEK293 reporter cells when their endogenous VKOR (and VKOR-like enzyme, VKORL) or FSP1 was knocked out. When VKOR was ablated (double-gene knockout, DKO)[23], these cells were unable to use $K_{epo}$ as a substrate to support VKD carboxylation (Fig. 3a). This is consistent with the notion that VKOR is the only enzyme for $K_{epo}$ reduction. However, VKR activity in these cells was

similar to wild-type HEK293 cells at higher vitamin K concentrations (Fig. 3b), suggesting that the enzyme responsible for vitamin K reduction in the DKO cells remained intact. The dramatically increased half-maximal effective concentration of vitamin K ($EC_{50}$) required for carboxylation in DKO cells (~360-fold increase, 2.36 μM vs 6.58 nM) (Fig. 3c) could result from either these cells having a lower efficiency for vitamin K reduction, or due to the cells losing the ability to recycle $K_{epo}$ by VKOR. Alternatively, when FSP1 was knocked out (FSP1 KO), the maximum carboxylation activity in the cell using either vitamin K or $K_{epo}$ (as a substrate) was considerably affected (decreased >80%) (Fig. 3a, b), suggesting that FSP1, and not VKOR, plays an essential role in vitamin K reduction. Compared with wild-type HEK293 cells, the $EC_{50}$ of vitamin K for the FSP1 KO cells increased only ~4-fold (27.0 nM vs 6.58 nM) (Fig. 3c), indicating that the recycling of $K_{epo}$ by VKOR in FSP1 KO cells make it efficient for VKD carboxylation, despite of its lower activity. Together, these results suggest that the dramatically increased $EC_{50}$ of vitamin K for VKD carboxylation in DKO cells results from the cells losing the ability to recycle $K_{epo}$ by VKOR, supporting the essential role of VKOR in VKD carboxylation as previously reported[24–26]. It is worth noting that the remaining carboxylation

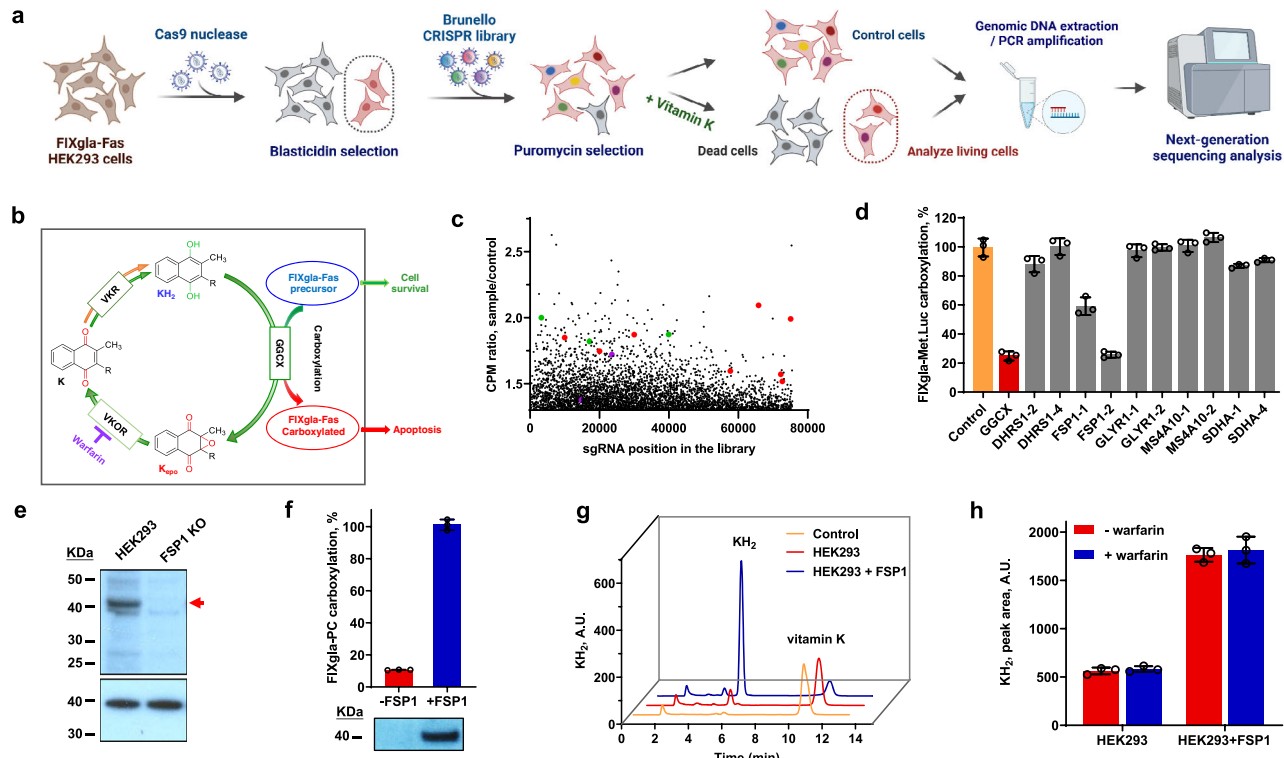

**Fig. 2 | Genome-wide CRISPR-Cas9 knockout screening of warfarin-resistant VKR. a** Schematic diagram of genome-wide loss-of-function screening using Brunello lentiviral library with FIXgla-Fas/HEK293 reporter cells. This figure was created with BioRender.com. **b** Scheme of VKD carboxylation of reporter protein FIXgla-Fas by the vitamin K redox cycle. Apoptosis of reporter cell occurs only when the reporter protein FIXgla-Fas is carboxylated. **c** Scatterplot of sgRNA enrichment after VKD apoptotic functional screening. Red dots, Fas-associated apoptosis pathway associated proteins; green dots, calcium-dependent proteins; purple dots, GGCX. **d** Cell-based validation of the candidate genes for vitamin K reduction in the presence of warfarin. Lentivirus containing Cas9 and selected sgRNA were transduced into FIXgla-*Met*.Luc/HEK293 reporter cells. After puromycin selection for 7 days, survival cells were incubated with 11 μM vitamin K and 5 μM warfarin for 24 h. Carboxylation efficiency of the reporter protein FIXgla-*Met*.Luc was determined by luminescence ELISA. VKD carboxylation efficiency of non-targeting sgRNA transfected cells was normalized to 100%. **e** Immunoblotting of HEK293 cells and these cells with their *fsp1* gene knocked out (FSP1 KO). Top panel: probed by anti-FSP1

antibody; bottom panel (loading control): probed by anti-GAPDH antibody. Full-length FSP1 is indicated by an arrowhead. **f** Effect of FSP1 knockout on VKR activity in HEK293 cells. FSP1 was knocked out (-FSP1) from FIXgla-PC/HEK293 or it was re-introduced back into the knockout cells (+FSP1) for VKR activity assay. The carboxylation activity of FSP1 transfected cells (+FSP1) was normalized to 100%. Bottom: Immunoblotting of FSP1 in the corresponding cells using anti-FSP1 as the primary antibody. **g** HPLC-based conventional VKR in vitro activity assay to determine the reduction of vitamin K to KH₂. Control: reaction buffer without cell lysate; HEK293, cell lysate of HEK293; HEK293 + FSP1: cell lysate of HEK293 over-expressing FSP1. A same number of HEK293 and HEK293 + FSP1 cells were used for the activity assay. **h** Warfarin inhibition of FSP1 reducing vitamin K to KH₂ by in vitro activity assay as described above (Fig. 2g). Final concentration of warfarin in the reaction mixture was 100 μM. Data are presented as mean ± SD of three independent experiments (n = 3) in Fig. 2d, h. Similar results were observed at least twice as shown in Fig. 2e, f.

activity observed in the FSP1 KO cells supports the existence of an alternative pathway for vitamin K reduction, which is also seen in FSP1 knockout cells when vitamin K was used as a ferroptosis protection agent[13].

Consistent with the data from our in vitro activity assay (Fig. 2h) and the results from FSP1 knockout mice[13], VKR activity in DKO cells (mainly from FSP1) was resistant to warfarin inhibition (Fig. 3d) strongly supported the notion that FSP1 was a warfarin-resistant VKR. However, when HEK293 cells had their *fsp1* gene knocked out (FSP1 KO), the residual VKR activity was in fact sensitive to warfarin inhibition to a similar inhibition potency as that of warfarin inhibition of VKOR reducing K_epo (Fig. 3d). This result appears to support the assumption that VKOR is responsible for the warfarin-sensitive vitamin K reduction[27]. Nevertheless, knocking out VKOR (and VKOR-like enzyme, VKORL) in the FSP1 KO cells (Triple-gene knockout, TKO, Supplementary Fig. 4a) did not eliminate VKR activity (Fig. 3e), suggesting that an additional enzyme was involved in vitamin K reduction; apart from FSP1 and VKOR. Again, knocking out VKOR significantly increased the EC₅₀ of the residual VKR activity, suggesting the importance of VKOR for recycling K_epo. In addition, the residual VKR activity in the TKO cells were no longer sensitive to warfarin inhibition

(Supplementary Fig. 4b). To further clarify this observation, we evaluated the capability of other NAD(P)H-dependent oxidoreductases for vitamin K reduction. Our results demonstrate that of the selected oxidoreductases, only NQO1 ((NAD(P)H-dependent quinone oxidoreductase 1) displays minor VKR activity (Fig. 3f) that is resistant to warfarin inhibition (Fig. 3g). Nevertheless, the overall contribution of this residual VKR activity to VKD carboxylation in TKO cells appears to be trivial when the relative activity was compared with wild-type HEK293 cells (Supplementary Fig. 4c).

It has been suggested that phylloquinone (vitamin K₁, referred to as vitamin K in this study) is the main form in the liver, which is responsible for VKD carboxylation of coagulation factors, while menaquinones (vitamin K₂) are mainly distributed in other tissues for extrahepatic carboxylation[28]. As FSP1 appears to have a low tissue specificity (Supplementary Fig. 5), it is reasonable to assume that FSP1 is responsible for both the hepatic and extrahepatic VKD carboxylation. The result of FSP1 not only reduces phylloquinone (Fig. 2g), but also the menaquinones (MK-4 and MK-7) to their corresponding hydroquinone forms (Fig. 3h, i), is consistent with previous work from our group showing that these vitamins can support VKD carboxylation[29], and a recent study showing that both forms of the

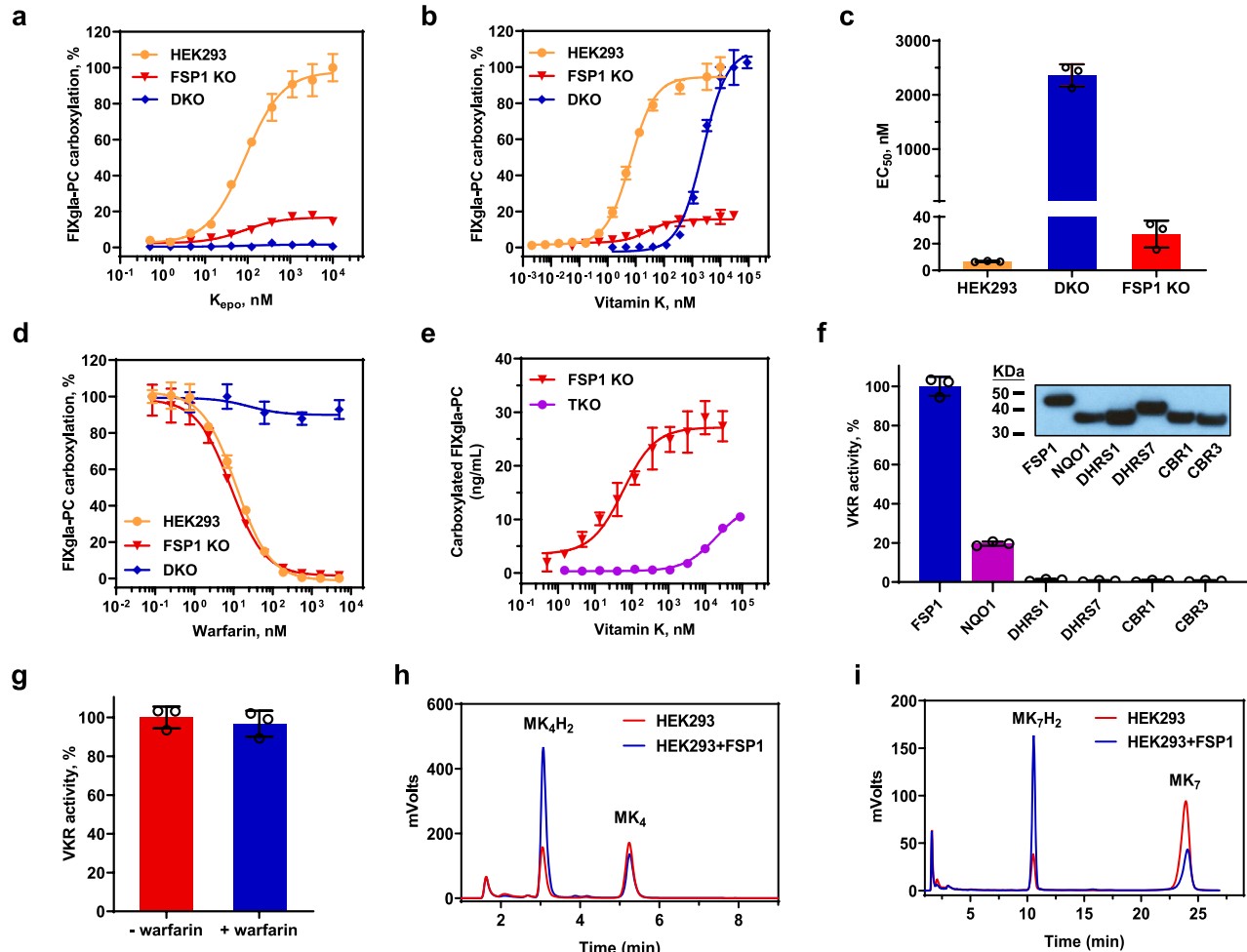

**Fig. 3 | Characterization of vitamin K reduction in different reporter cells. a** $K_{epo}$ and **b** vitamin K concentration titrations in different reporter cells. Reporter cells were incubated with increasing concentrations of $K_{epo}$ or vitamin K for 24 h. VKD carboxylation of the reporter-protein FIXgla-PC were determined using ELISA. Carboxylation activity of HEK293 cells at the highest $K_{epo}$ or vitamin K concentration was normalized to 100%. HEK293: FIXgla-PC/HEK293 reporter cell; FSP1 KO: FIXgla-PC/HEK293 cells with their endogenous *fsp1* gene knocked out; DKO: FIXgla-PC/HEK293 with their endogenous *vkor* and *vkorl* genes knocked out. **c** Half-maximal effective concentration of vitamin K ($EC_{50}$) determined in HEK293, DKO, and FSP1 KO reporter cells from Fig. 3b. **d** Warfarin inhibition of VKD carboxylation in DKO and FSP1 KO cells with vitamin K as the substrate, and in FIXgla-PC/HEK293 cells with $K_{epo}$ as the substrate. Increasing concentrations of warfarin with 11 µM vitamin K or 5 µM $K_{epo}$ were incubated with the corresponding reporter cells for 24 h for VKD carboxylation activity assay. Carboxylation activity of each cell line without warfarin was normalized as 100%. **e** VKD carboxylation efficiency in FSP1 KO and TKO cells using vitamin K as the substrate. Carboxylation activity was presented as concentration of carboxylated reporter-protein in the cell culture medium (ng/mL) determined by ELISA. **f** HPLC-based conventional VKR in vitro activity assay of the selected NAD(P)H-dependent oxidoreductases in TKO cells. Candidate proteins were transiently expressed in TKO cells for 48 h and VKR activity of reducing vitamin K to $KH_2$ was determined. VKR activity of FSP1 was normalized to 100%. Insert: Immunoblotting of the expression of the tested oxidoreductases and similar immunoblotting results were observed at least twice. DHRS, dehydrogenase/reductase SDR family protein; CBR, carbonyl reductase. (**g**) Warfarin inhibition of NQO1 reducing vitamin K to $KH_2$ by HPLC-based VKR in vitro activity assay. Final concentration of warfarin in the reaction mixture was 100 µM. **h**, **i** reduction of menaquinone-4 (MK4) (**h**) and menaquinone-7 (MK7) (**i**) by FSP1 determined by monitoring the production of their corresponding hydroquinone by HPLC-based VKR in vitro activity assay. HEK293: cell lysate of HEK293 cells; HEK293 + FSP1, cell lysate of HEK293 cells overexpressing FSP1. A same number of HEK293 and HEK293 + FSP1 cells were used for the activity assay. Data are presented as mean ± SD of three independent experiments ($n = 3$) in Fig. 3a–g.

vitamins supporting ferroptosis suppression[13]. These results taken together suggest that VKOR is essential for recycling $K_{epo}$, and FSP1 is the enzyme mediating warfarin-resistant vitamin K reduction to underpin VKD carboxylation. The alternative vitamin K reduction pathways, including VKOR and possibly NQO1, only contribute negligibly to VKD carboxylation in our test conditions.

### Diverse functions and subcellular localizations of FSP1

FSP1 was initially named as AIFM2, a mitochondria protein associated with apoptosis[30,31]. Recent studies show that FSP1, when recruited to the plasma membrane, functions as an oxidoreductase reducing ubiquinone (coenzyme Q10, $CoQ_{10}$) to ubiquinol ($CoQ_{10}H_2$), which acts as a lipophilic radical-trapping antioxidant to inhibit ferroptosis, and therefore it was renamed as ferroptosis suppressor protein 1 (FSP1)[32,33]. While others purported FSP1 as a lipid droplet-associated protein highly enriched in brown adipose tissue, and is a reduced nicotinamide adenine dinucleotide (NADH) oxidase to regenerate cytosolic $NAD^+$ to promote thermogenesis[34]. The findings herein suggest that FSP1 reduces vitamin K to support VKD carboxylation in the endoplasmic reticulum (ER). Consistent with previous findings[32], our examination of the subcellular localization of FSP1 shows that it is localized to unspecified perinuclear membrane compartments that overlap with ER, Golgi, and mitochondria markers (Fig. 4a). Thus, the multiple subcellular localization of FSP1, and its substrate promiscuity ($CoQ_{10}$ and vitamin K), like other NAD(P)H-dependent oxidoreductases, supports a diverse range of its biological functions.

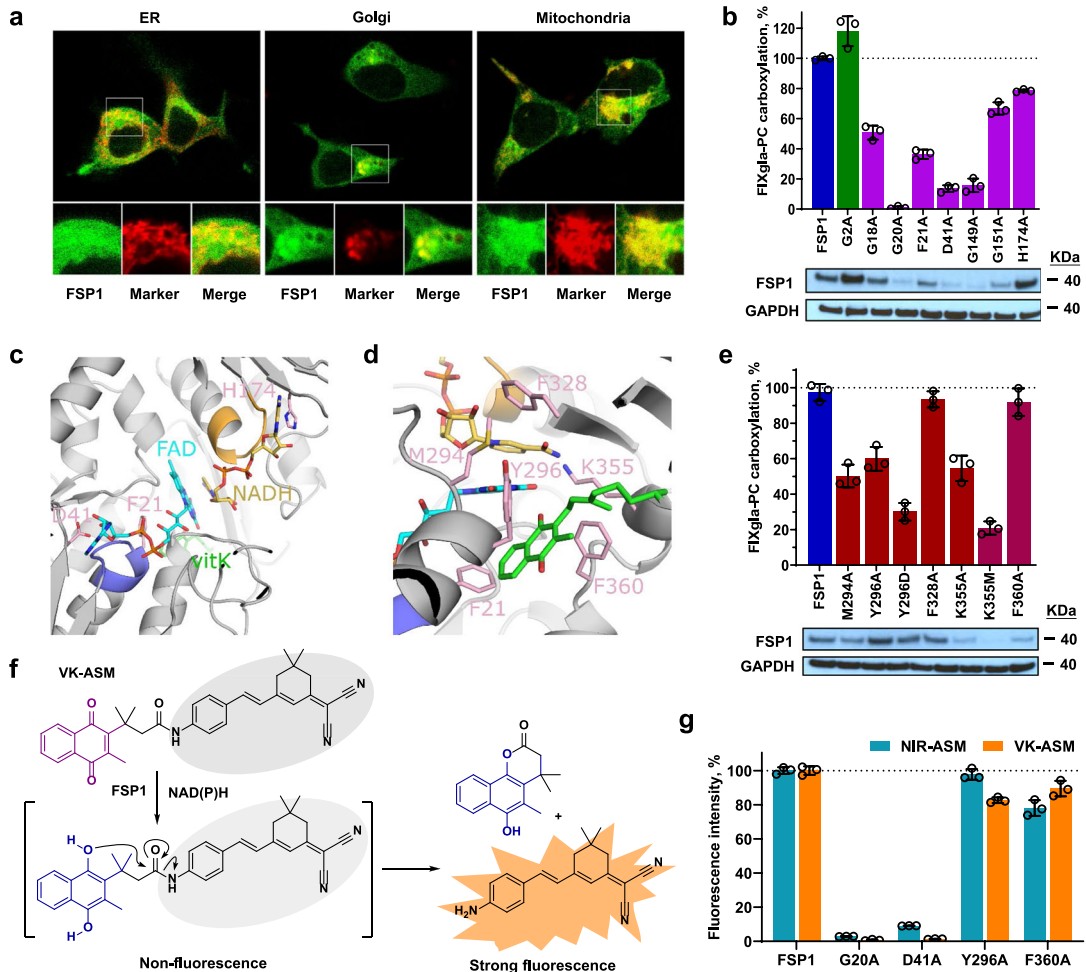

**Fig. 4 | Localization and characterization of FSP1 as vitamin K reductase.**
**a** Fluorescence confocal microscope imaging of FSP1-sfGFP fusion (green) with cell organelle markers (red) of ER, mitochondrion, and Golgi. FSP1-sfGFP and the cell organelle marker were transiently co-transfected into HEK293 cells for 48 h. **b** Effect of FSP1 myristylation site mutation (G2A) and mutations of residue responsible for the dinucleotides (NDAH and FAD) binding on VKR activity. Wild-type FSP1 or its mutants was transiently expressed in TKO cells for VKR activity assay. Wild-type FSP1 activity was normalized to 100% (indicated by dotted line). Bottom: Immunoblotting of FSP1, its mutants, and GAPDH (loading control) in the corresponding cells. **c** AlphaFold model of FSP1 (gray) with FAD (cyan), NADH (gold) and vitamin K (green) based on ligand positions in the yeast NDH-2 enzyme Ndi1 (PDBcode 4G73). The GxGxxG motifs are colored purple (Gly18-Gly23) and orange (Gly149-Gly154) corresponding to interactions with the nucleotide motifs of FAD and NADH respectively. Residues Asp41 and His174 located near the ribose hydroxyls from the two nucleotides are colored pink as is Phe21 from the first GxGxxG region. **d** FSP1 residues surrounding the isoalloxazine of FAD,

nicotinamide of NADH and quinone of vitamin K in the model that were mutated in this study are colored pink. **e** Effect on FSP1 activity due to mutations surrounding the proposed vitamin K binding site. VKR activity was determined as described above (Fig. 4b). Bottom: Immunoblotting of FSP1, its mutants, and GAPDH (loading control) in the corresponding cells. **f** Proposed activation mechanism of the activity-based fluorescent probe of vitamin K (VK-ASM) towards FSP1. The probe is inactive until the vitamin K moiety is reduced to hydroquinone, and the rearrangement of the reduced intermediated releases the strong fluorescent tag. **g** Effect of FSP1 mutations on the reduction of vitamin K and $CoQ_{10}$. Wild-type FSP1 and its mutants were transiently expressed in TKO cells for forty-eight hours, cell lysate was used for activity assay using the activity-based fluorescent probe of vitamin K (VK-ASM) and $CoQ_{10}$ (NIR-ASM) as the substrate. Fluorescence intensity of the wild-type FSP1 for each probe was normalized to 100% (indicated by the dotted line). Data are presented as mean ± SD of three independent experiments ($n = 3$) in Fig. 4b, e, g. Similar results were observed at least twice as shown in Fig. 4b, e.

Mutating the myristoylation site (G2A) of FSP1 impaired its ferroptosis suppression activity[13,32,33], but not VKD carboxylation activity (Fig. 4b), suggesting that this mutation does not affect the catalytic activity of the enzyme for quinone reduction. Sequence analysis suggests that FSP1 contains two Rossmann-fold motifs[35], the canonical motif for dinucleotide cofactor binding (Supplementary Fig. 6). FSP1 was suggested to be a type II NADH dehydrogenases (NDH-2)[36] that reduces quinone via shuttle reducing equivalents from NADH to flavin adenine dinucleotide (FAD) or flavin mononucleotide (FMN) through a ternary complex with both NAD(H) and quinone bound or a ping-pong mechanism, whereby the NAD+ has dissociated prior to quinone binding[37–39]. FAD, NADH, and vitamin K were positioned into the predicted AlphaFold model of FSP1 based on cofactor and ligand positions

from the crystal structure of yeast Ndi1 (Fig. 4c). Residues Gly18 to Gly23 and Gly149 to Gly154 form the consensus GxGxxG motifs of the Rossmann-fold that contribute to binding of the nucleotides of FAD and NADH, respectively[40,41].

Mutating residues Gly18, Phe21, Gly149, and Gly151 decreased FSP1 activity by 35-85%, while the G20A mutant abolished FSP1 activity (Fig. 4b), which is consistent with previous observations from other NADH-dependent flavin-oxidoreductase[40,42], thus supporting the importance of these GxGxxG motifs for dinucleotide cofactor binding. Based on the ternary complex model, Asp41 and His174 are specifically placed to interact with the ribose moieties of the FAD and NADH nucleotides, respectively (Fig. 4c). While the H174A mutant displayed modest reduction in carboxylation activity, the D41A mutant reduced

carboxylation by ~80%. All these residues are fully conserved across 30 species (Supplementary Fig. 7) supporting the relative positioning of the FAD and NADH nucleotides in the model. Additionally, western blot analysis shows that mutating the proposed dinucleotide cofactor binding residues significantly decreased protein expression levels (Fig. 4b, bottom), which is consistent with a previous study suggesting that these residues are essential for flavoprotein folding and/or stability[43].

Based on the above model, sidechains from residues Met294, Phe328, and Lys355 are in close proximity to the isoalloxazine ring and nicotinamide moieties of the FAD and NADH, respectively (Fig. 4d). Although the F328A mutant did not impact carboxylation activity, M294A and K355A displayed a ~50% activity reduction while K355M displayed a ~80% activity reduction (Fig. 4e), supporting roles in functional group positioning for residues Met294 and Lys355. The placement of vitamin K in the model indicates that the naphthoquinone of vitamin K could be positioned between Tyr296 and Phe360, with Phe21 lining the inner side of the pocket (Fig. 4d). While the F360A mutant displayed wild-type carboxylation activity, Y296A and Y296D reduced the activity by 40% and 70%, respectively (Fig. 4e). The 60% activity reduction observed for the F21A mutant could be due to a disruption of FAD binding (within the GxGxxG motif) and/or vitamin K binding. Unexpectedly, our immunoblotting results indicate that mutations at residue Lys355 and Phe360 dramatically decreased protein expression levels (Fig. 4e, bottom), despite their significant activity. We suspect that this is due to the FSP1 antibody we used in the immunoblotting recognizing residues near the C-terminus of the protein, thus mutations at Lys355 and Phe360 could disrupt antibody recognition. To test this hypothesis, we fused a His-tag at the C-terminus of FSP1 mutations that displayed significantly decreased protein levels in the immunoblot. Results in Supplementary Fig. 8 show that when the blot was probed by His-tag antibody, the Lys355 and Phe360 mutants have a similar protein level as the wild-type FSP1, but not the Gly20, Gly149, and Gly151 mutants, suggesting that the lower protein levels of Lys355 and Phe360 mutants in the immunoblotting analysis (Fig. 4e, bottom) was due to the FSP1 antibody recognition.

The AlphaFold model of FSP1 displays large deviations in the C-terminus from that of NDH-2 crystal structures of yeast Ndi1 and AIF[41,44] (Supplementary Fig. 9a–c). This difference in ternary structure results in greater uncertainty with regards to the position of Phe360 and/or the alkyl chain of the vitamin K. Nevertheless, mutations of the residues surrounding the functional groups of the FAD, NADH and vitamin K, decreased carboxylation activity, emphasizing the positional importance of these groups. In addition, residues Phe21, Met294, Tyr296, and Lys355 are conserved across the 30 species (Supplementary Fig 7). Mutations in these residues displayed greater impact on carboxylation activity than mutations to Phe328 and Phe360, which are less conserved.

As the diverse functions of FSP1 result from its substrate promiscuity (CoQ$_{10}$ and vitamin K), we compared the effect of select residues on the reduction of vitamin K and CoQ$_{10}$. To directly compare the enzymatic activity of FSP1 to both substrates, we synthesized a de novo activity-based fluorescent probe of vitamin K (VK-ASM) (Fig. 4f and Supplementary Fig. 10), encouraged by a similar fluorescent probe of CoQ$_{10}$ (i.e., NIR-ASM)[45,46] (Supplementary Fig. 11a). The vitamin K specific probe utilized herein becomes fluorescent on reduction of the quinone moiety to the transient hydroquinone, which is followed by a subsequent rearrangement to release the strongly fluorescent tag (moiety) (Supplementary Fig. 11b). Deployment of the fluorescent probe revealed that mutating the residues corresponding to the cofactor and quinone substrate binding site have a similar effect on both vitamin K and CoQ$_{10}$ reduction (Fig. 4g and Supplementary Fig. 12). Thus, suggesting that both substrates are reduced via the same mechanism.

## Selectively sensitizing CoQ$_{10}$-linked ferroptosis

Previous studies shown that FSP1 suppresses ferroptosis through the reduction of CoQ$_{10}$ in the plasma membrane[32,33], while dihydroorotate dehydrogenase (DHODH) reduces CoQ$_{10}$ in mitochondria for ferroptosis suppression[47] (Fig. 5a). Although both FSP1 and DHODH can reduce CoQ$_{10}$ to CoQ$_{10}$H$_2$ to suppress ferroptosis, only FSP1 exhibits dramatic activity for vitamin K reduction to support VKD carboxylation (Fig. 5b). The inability of DHODH for supporting VKD carboxylation could be due to DHODH being a mitochondrial inner membrane protein while VKD carboxylation occurs in the ER lumen. Intriguingly, the FSP1 inhibitor iFSP1, which robustly sensitized cancer cells to RSL-3-induced ferroptosis[32], and its competitive inhibitor HQNO[36,38], also displayed significant inhibition on VKD carboxylation (Fig. 5c). However, the DHODH potent inhibitors, Leflunomide (LFM) and Brequinar (BRQ), which suppressed the tumor growth of GPX4$^{low}$ xenografts[47], have a negligible effect on VKD carboxylation (Fig. 5d). As these inhibitors occupy the same binding site as ubiquinone[38,48], one explanation for these divergent results is that the ubiquinone binding pockets of FSP1 and DHODH are different, which could result from their distinct reaction mechanisms for ubiquinone reduction. FSP1 uses NADH as a cofactor to shuttle reducing equivalents to FAD for ubiquinone reduction, while DHODH obtains reducing equivalents from oxidizing DHO and transferring electrons to FMN for ubiquinone reduction.

While HQNO is a competitive inhibitor of NDH-2, such as FSP1[36,38], a Lineweaver-Burk plot of the experimental data from our cell-based inhibition study shows that plotted lines of HQNO inhibiting FSP1 intersect to the left of the $y$-axis and below the $x$-axis (Fig. 5e), suggesting a mixed inhibition of HQNO to vitamin K reduction in HEK293 cells. This observation is consistent with the above result showing that multiple enzymes are involved in vitamin K reduction. Nevertheless, it is clear that inhibition of FSP1 for sensitizing ferroptosis will also inhibit VKD carboxylation. Therefore, targeting the FSP1-CoQ$_{10}$-NADH ferroptosis defense pathway, but not the DHODH-CoQ$_{10}$-DHO pathway, could negatively interfere with VKD carboxylation.

## Discussion

VKD carboxylation relies on the redox cycling of vitamin K. In vitro studies suggest that vitamin K is reduced to KH$_2$, the active form of the vitamin for VKD carboxylation, via two pathways – a warfarin-sensitive DTT-dependent pathway and a warfarin-resistant NAD(P)H-dependent pathway[27,49]. VKOR was proposed to be responsible for the warfarin-sensitive pathway[50], and the enzyme for the warfarin-resistant (or the antidotal) pathway was proposed to be NQO1[51]. However, we have previously shown that high doses of vitamin K sufficiently rescued warfarin poisoning in both wild-type and NQO1-deficient mice, and the warfarin-resistant VKR activity (i.e., distinct from VKOR and NQO1) were detected in both wild-type and NQO1-knockout mice, suggesting that NQO1 was not the warfarin-resistant VKR[52]. Whether VKOR is sufficient for the two-step reduction of K$_{epo}$ to KH$_2$, and whether the antidotal effect of vitamin K is due to the competition of vitamin K with warfarin for binding to VKOR, or even due to the existence of a warfarin-resistant VKR (antidotal enzyme) that bypasses dysfunctional VKOR, have all remained elusive.

To clarify these issues, we employed a genome-wide CRISPR-Cas9 knockout screening for the identification of VKR. Phenotypic screening of a genome-wide pooled library requires survival and growth phenotype difference of the library infected cells given a specific selective pressure. Screening by positive selection has the highest signal-to-noise ratio compared to other types of screening[17]. However, it relies on the enrichment of sgRNAs for genetic perturbation, which in turn produces the screening phenotype via cell proliferation. The VKD apoptotic reporter cell line described in this study now makes it possible to screen for any enzymes that are associated with VKD carboxylation. Our results reinforce recent findings[13] that FSP1 is responsible for the warfarin-resistant vitamin K reduction.

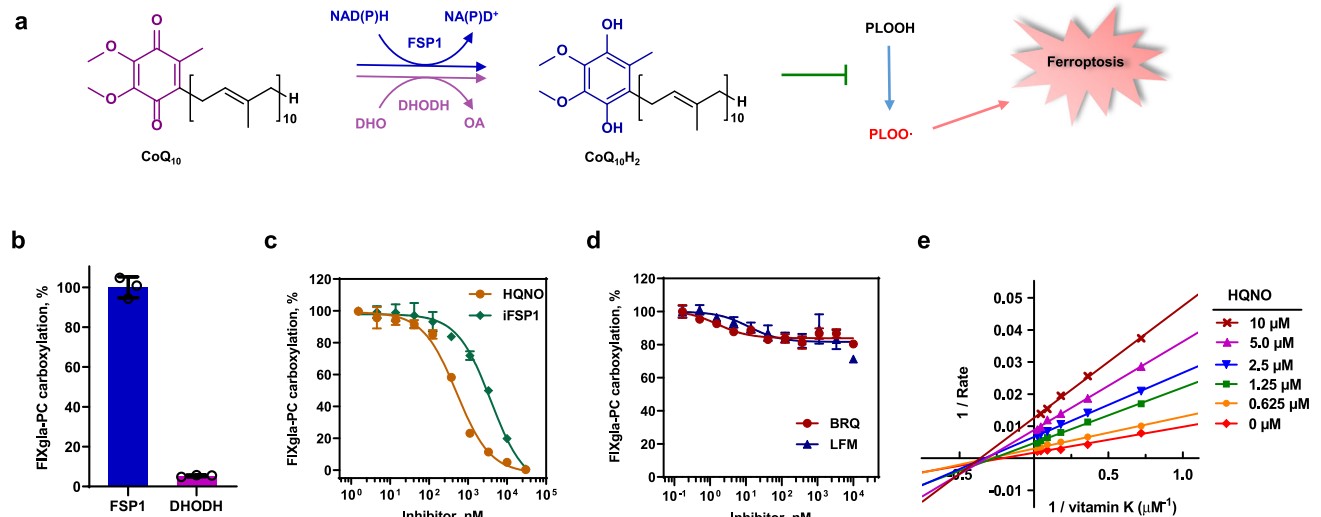

**Fig. 5 | Effect of the CoQ10-associated ferroptosis suppression pathways on VKD carboxylation. a** Scheme of the FSP1-CoQ10-NADH and DHODH-CoQ10-DHO pathways for ferroptosis suppression. CoQ10 can be reduced to CoQ10H2 by either FSP1 or DHODH using the reducing equivalent from NADH or DHO, respectively. Reduced CoQ10 traps lipophilic radicals to suppress ferroptosis. NADH reduced nicotinamide adenine dinucleotide, NAD+ oxidized nicotinamide adenine dinucleotide, DHO dihydroorotate, OA orotic acid, PLOOH phospholipid hydroperoxides, PLOO· phospholipid peroxyl radical. **b** Cell-based VKR activity of FSP1 and DHODH. FSP1 and DHODH were transiently expressed in TKO reporter cells for 24 h for carboxylation activity assay. Wild-type FSP1 activity was normalized to 100%. **c** Effect of FSP1 inhibitors on VKD carboxylation. Increasing concentrations of FSP1 inhibitors, HQNO or iFSP1, with 11 μM vitamin K were incubated with DKO reporter cells for 24 h for VKD carboxylation activity assay. Carboxylation activity of cells incubated with 11 μM vitamin K without inhibitor was normalized as 100%. **d** Effect DHODH inhibitors on VKD carboxylation. Increasing concentrations of DHODH inhibitors, BRQ and LFM, with 11 μM vitamin K were incubated with DKO reporter cells for 24 h for VKD carboxylation activity assay. Carboxylation activity of cells incubated with 11 μM vitamin K without inhibitor was normalized as 100%. **e** Lineweaver–Burk plot of VKR activity in DKO cells as a function of vitamin K concentration in the absence and presence of HQNO. DKO reporter cells were cultured with increasing concentrations of vitamin K containing 0, 0.625, 1.25, 2.5, 5.0, and 10 μM HQNO for the activity assay. Data were plotted using Graphpad software. Data are presented as mean ± SD of three independent experiments (n = 3) in Fig. 5b–d.

Characterization of VKOR and FSP1 knockout HEK293 cells suggests that VKOR is essential for K$_{epo}$ reduction and efficient recycling of vitamin K, while FSP1 is mainly responsible for the warfarin-resistant vitamin K reduction to support VKD carboxylation. It is worth noting that in addition to FSP1, other enzymes also partially contribute to vitamin K reduction to support carboxylation (Fig. 3). This result and results from FSP1-knockout mice[34,53,54] suggest that, despite the significantly low level, the alternative vitamin K reduction pathway is sufficient to maintain a homeostatic balance of blood coagulation, unless it is challenged by warfarin[13]. Therefore, FSP1 is an ideal therapeutic target for thrombosis without bleeding risk.

FSP1 is a flavoprotein localized to multiple subcellular organelles. We identified two Rossmann-fold motifs in FSP1 that are responsible for FAD and NADH binding. Mutating glycine residues in the conserved GxGxxG sequence within the Rossmann-fold motif significantly affected vitamin K reduction activity in cells. Using a fluorescence probes of vitamin K and ubiquinone, we have shown that FSP1 utilizes the same binding pocket for vitamin K and ubiquinone reduction. It is worth noting that, different from VKOR that reduces vitamin K by the active site free cysteines (Cys132 and Cys135) which are oxidized to a disulfide after vitamin K reduction[21,55], FSP1 does not have "active site residues" that are directly involved in vitamin K reduction. Instead, it likely shuttles reducing equivalents from NADH to FAD where reduced FAD reduces vitamin K. Therefore, we assume that inactivation of the G20A mutant (and reduced activity by other GxGxxG mutants in this study) for vitamin K reduction could be due to a lack of enzyme binding to FAD and/or affecting the stability of the enzyme as previously reported[43].

FSP1 was recently renamed due to its function in ferroptosis suppression[32,33]. Ferroptosis is a regulated non-apoptotic cell death characterized by the iron-dependent excessive lipid peroxidation of cellular membranes[56]. It has been implicated in a broad range of physiological and pathological contexts[57,58]. Several endogenous ferroptosis surveillance pathways have been identified. Conspicuously, two of these are associated with reduced CoQ10 protection against lipid peroxidation, the FSP1-CoQ10-NADH pathway[32,33] and the DHODH-CoQ10-DHO pathway[47] (Fig. 5a). Suppression of the CoQ10-related defense system appears to be useful for sensitizing ferroptosis to kill cancer cells. For example, FSP1 is abundantly expressed in most tumor cell lines and pharmacological inhibition of FSP1 by small-molecule inhibitors robustly sensitized cancer cells to RSL3-induced ferroptosis[32,59]. Additionally, genetic deletion of FSP1 significantly reduced tumor growth in a xenograft mouse model[33]. Furthermore, combined with glutathione peroxidase 4 (GPX4)-knockdown, inhibition of DHODH by its inhibitor, brequinar, markedly suppressed xenograft tumor growth mediated by ferroptosis[47]. Results from this study imply that targeting the FSP1 pathway, but not the DHODH pathway, for sensitizing ferroptosis to kill cancer cells could also impact VKD carboxylation. Importantly, unlike warfarin poisoning, which can be rescued by administering large doses of vitamin K, toxicity caused by inhibition of FSP1 may not be reversed as FSP1 is the enzyme that directly provides reduced vitamin K for carboxylation in the vitamin K cycle. As reduced vitamin K not only serves as a cofactor for VKD carboxylation, it also functions as a mitochondrial electron carrier[60], attempts to therapeutically manipulate vitamin K reduction could have far reaching impact, especially considering vitamin K reductase was established as a multi-cell organelle protein with substrate promiscuity i.e., it can reduce both vitamin K and ubiquinone.

## Methods
### Reagents and antibodies
All chemicals including activity-based fluorescent probe of ubiquinone (NIR-ASM, #SCT072), menaquinone-4 (MK-4, #V9378), menaquinone-7 (MK-7, #1381119), NADH (#10128023001), and cOmplete EDTA-free

protease inhibitor cocktail (#4693132001) were purchased from Sigma-Aldrich. HQNO (2-*n*-heptyl-4-hydroxyquinoline-*N*-oxide, #15159), iFSP1 (#29483), Brequinar (BRQ, #36183) and Leflunomide (LFN, #14860) were sourced from Cayman Chemical. Vitamin K (10 mg/mL, injectable emulsion, #NDC 0409-9158) for cell culture was obtained from Novaplus. Vitamin K 2,3-epoxide was prepared by oxidizing vitamin K (Sigma-Aldrich, viscous liquid) by hydrogen peroxide in the presence of sodium carbonate[61]. Xfect transfection reagent (#631318) was from TaKaRa. Cell counting kit CCK-8 kit (#CK04-011) was from Dojindo Molecular Technologies, Inc. Luciferase substrate native Coelenterazin (CTZ, #3031) was obtained from NanoLight Technologies. pcDNA 3.4 TOPO TA cloning kit (#A14697) and ECL western blotting detection kit (#WP20005) were from ThermoFisher Scientific. ABTS (2,2'-Azino-bis(3-ethylbenzothiazoline-6-sulfonic acid) diammonium salt) peroxidase substrate kit (#5120-0041) for ELISA was from KPL Inc. QuickChange site-directed mutagenesis kit (#200524) was from Agilent Technologies, Inc. Accutase cell detach buffer (#AT-104) was obtained from Innovative Cell Technologies, Inc.

Mammalian expression vector pcDNA3.1/hygro(+) (#V87020) and phCMV 1 (#P003100) were from Invitrogen and Genlantis, respectively. Human Brunello CRISPR knockout pooled library was a gift from David Root and John Doench (Addgene #73178)[18]. The fluorescent protein-tagged marker of the endoplasmic reticulum (mCherry-Sec61-N-18), mitochondrial (mCherry-TOMM20-N-10), and Golgi apparatus (pmScarlet_Giantin_C1) were gifts from Michael Davidson (Addgene plasmid # 55130 and #55146) and Dorus Gadella (Addgene plasmid # 85048), respectively. The sgRNA cloning vector of pX330-U6-Chimeric-BB-CBh-hSpCas9, LentiCRISPR v2, and lentiCas9-Blast were gifts from Feng Zhang (Addgene plasmid # 42230, #52961, and #52962, respectively). Lentiviral packaging plasmids pMD2.G and psPAX were gifts from Didier Trono (Addgene plasmid #12259 and #12260, respectively).

Mouse anti-carboxylated FIX gla domain monoclonal antibody (#GMA-001, diluted 1:1000) was obtained from Green Mountain Antibodies. Mouse anti-GAPDH monoclonal antibody (#60004-1-Ig, diluted 1:6000) and HRP-conjugated anti-His-Tag monoclonal antibody (#HRP-66005, diluted 1:2000) were from Proteintech Group, Inc. Affinity purified sheep anti-human Protein C IgG (#SAPC-AP, diluted 1:4000) and its horseradish peroxidase conjugate (#SAPC-HRP, diluted 1:1000) were from Affinity Biologicals Inc. Mouse anti-HPC4 monoclonal antibody was a gift from Dr. Charles Esmon (Oklahoma Medical Research Foundation, Oklahoma City, OK, diluted 1:2000). Mouse anti-Fas monoclonal antibody (#8023, diluted 1:1,000), mouse anti-Caspase-8 monoclonal antibody (#9746, diluted 1:1000), and rabbit anti-AIFM2/FSP1 polyclonal antibody (#24972, diluted 1:1000) (for endogenous FSP1 detection) were obtained from Cell Signaling Technology. Rabbit anti-AIFM2 monoclonal antibody (#ZRB1527, diluted 1:1000) (for overexpressed FSP1 detection) was obtained from Sigma-Aldrich.

## Cell lines and cell culture

Human embryonic kidney 293 (HEK293, #CRL-1573) cells were obtained from ATCC and were cultured in DMEM/F-12 (Dulbecco's Modified Eagle Medium/Nutrient Mixture F-12, Gibco #11330-032) medium with 10% fetal bovine serum (FBS, Sigma-Aldrich #F2442) and 1x penicillin streptomycin (Corning #30-002 CL) at 37 °C in a 5% CO2 incubator. HEK293 cells stably expressing of the reporter protein FIXgla-Fas or FIXgla-*Met*.Luc (*Metridia* luciferase) were obtained by transfecting of HEK293 cells with corresponding chimeric reporter protein constructs in pcDNA3.1 plasmid DNA using Xfect transfection reagent, according to the manufacturer's instruction. After selection with 300 μg/ml hygromycin, surviving colonies were picked and screened for high expression of FIXgla-Fas or FIXgla-*Met*.Luc using immunoblotting analysis or luciferase activity. The colony with the highest FIXgla-Fas production was selected as the stable cell line for genome-wide CRISPR-Cas9 knockout loss-of-function screening. The maintenance of FIXgla-Fas/HEK293 cells included 2 μM warfarin in the cell culture medium to inactive VKD carboxylation resulting from the residual vitamin K in the medium. HEK293 cells stably expressing the reporter protein FIXgla-PC and these cells with their endogenous VKOR (and VKOR-like enzyme, VKORL) knocked out (DKO) were obtained, as previously described[19,23]. FSP1 knockout in FIXgla-PC/HEK293 cells (FSP1 KO) and in DKO cells (TKO) were obtained by CRISPR-Cas9-mediate genome editing[29]. The sgRNA targeting exon 4 (CCAGCGCTCACGGTTCATCG) of *fsp1* was cloned into sgRNA cloning vector pX330 for genome editing. Single colonies of *fsp1* knockout were obtained by functional screening and genomic DNA from the single colony was extracted and used as a template for PCR amplification of the sgRNA targeting region using the following forward and reverse primer. Forward primer 5'–3': GCAGGCTTAGATCCACA ATCTTCCACAG. Reverse primer 5'–3': CCTTTCCTTGAGAAAGGACA CACCCTGAG. The resulted PCR product was cloned into pcDNA 3.4 TOPO TA cloning vector and 10 positive colonies from each knockout cell line were picked up for sequencing analysis.

## DNA manipulations and plasmids construction

The cDNAs of Fas (Clone ID BC012479), FSP1 (Clone ID BC023601), DHODH (Clone ID BC065245), NQO1 (Clone ID BC007659), DHRS1 (Clone ID BC014057), DHRS7 (Clone ID BC000637), CBR1 (Clone ID BC002511), and CBR3 (Clone ID BC002812) were obtained from Transomic Technologies, Inc. For expression of these genes in HEK293 cells, the cDNA of the individual gene was cloned into phCMV 1 mammalian expression vector. To verify of the expression, we fused a 8xHis-Tag at the C-terminus of each gene for immunoblotting detection. The FSP1 mutations were created by QuickChange site-directed mutagenesis using wild-type FSP1 in pcDNA3.1 vector as the template. For confocal fluorescence imaging of FSP1, sfGFP was fused to the C-terminus of FSP1 with a flexible linker of GGSGG using overlap PCR. The chimeric reporter protein of FIXgla-Fas with a HPC4 tag (EDQVDPRLIDGK) and a flexible linker GGSGG following FIXgla was obtained by overlap PCR and cloned into pcDNA3.1 vector. The FIXgla portion includes the signal peptide, propeptide, and gla domain of FIX. To validate the candidate genes from CRISPR-Cas9 knockout screening, the top two sgRNAs for each candidate gene, one positive control (sgRNA targeting GGCX), and one negative control (non-targeting sgRNA) were individually cloned into the lentivirial plasmid backbone lentiCRISPR v2 which contains the Cas9 component.

The nucleotide sequences of all the constructs were verified by DNA sequencing at Eton Bioscience Inc. (Research Triangle Park, NC).

## Fluorescence confocal microscopy

To determine the expression and carboxylation of FIXgla-Fas in live cells, immunofluorescence confocal microscopy imaging was used, as previously described[62]. In brief, FIXgla-Fas/HEK293 cells were incubated with 11 μM vitamin K or 5 μM warfarin for 12 h. Cells were washed with phosphate buffered saline solution (PBS) and detached from culture flask using a mild cell detachment buffer Accutase to protect the extracellular portion of the cell membrane proteins. Cells were then co-immunostained with FITC-labeled anti-carboxylated FIXgla antibody and APC-labeled anti-HPC4 antibody in TBS buffer containing 2 mM CaCl₂. Unreacted antibodies were washed off before fluorescence confocal imaging.

The subcellular localization of FSP1 were examined by co-expression of FSP1-sfGFP fusion with one of the corresponding red fluorescent protein-tagged cell organelle marker protein in HEK293 cells on coverslips. Forty-eight hours post transfection, cells were washed with PBS and fluorescence confocal microscopy imaging was performed on a Zeiss LSM710 confocal laser scanning microscope[63].

## Cell viability assay

The sensitivity of FIXgla-Fas/HEK293 cells to vitamin K induced apoptosis was evaluated by cell viability assay. FIXgla-Fas/HEK293 cells cultured in warfarin-containing medium were seeded to a 96-well plate. Next day, cell culture medium was replaced by fresh medium containing increasing concentrations of vitamin K. After 24 h incubation, cell viability was determined using a CCK-8 kit following the manufacturer's instructions.

## Immunoblotting

Western blot analysis of cell lysates was performed as previously described[29]. His-Tag proteins were probed directly by HRP conjugated anti-His-Tag antibody. Carboxylated FIXgla, HPC4 tag, Fas, and caspase8 were probed by their corresponding antibodies as the primary antibody, and HRP-conjugated goat anti-mouse or anti-rabbit IgG as the secondary antibody accordingly. Protein bands were visualized by ECL western blotting detection kit.

## Genome-wide CRISPR-Cas9 knockout screening

The Brunello CRISPR knockout pooled library[18], which contains 76,441 single guide RNAs (sgRNAs) targeting 19,114 genes in the human genome, was used for genome-wide loss-of-function screening. Lentivirus of the Brunello CRISPR library was produced in HEK293T cells by co-transfection of the sgRNA pooled library in LentiCRISPRv2 plasmid together with pMD2.G and psPAX lentiviral helper plasmids using Xfect as the transfection reagent in T225 flasks[17]. Supernatants were collected 48 h post-transfection and filtered with 0.45 μm filter. Lentiviral titer was determined using FIXgla-Fas/HEK293 reporter cells. Before transducing the Brunello lentivirial library, Cas9 nuclease was stably expressed in FIXgla-Fas/HEK293 reporter cells by lentiviral transduction with lentiCas9-Blast[64] and subsequently selected using 5 μg/ml blasticidin and expanded for 7 days.

The Brunello lentivirial sgRNA library was then delivered into FIXgla-Fas/HEK293-Cas9 cells by spinfection in 12-well plates at 1000×*g* for 2 h at 33 °C. To ensure that most cells receive only one genetic perturbation, four 12-well plates at a cell density of $3 \times 10^6$ cells in 2 ml of DMEM/F12 medium per well were used for library transduction at a multiplicity of infection (MOI) of 0.3 to achieve a coverage of ~565 cells expressing each sgRNA. Twenty-four hours post-spinfection, cells were trypsinized and transferred into eight 245 mm×245 mm cell culture dishes in DMEM/F12 medium containing 1.5 μg/ml puromycin. Three days later, survival cells were washed with PBS, trypsinized, and seeded back to eight 245 mm×245 mm culture dishes with fresh media containing 1.5 μg/ml puromycin and expanded for additional three days prior to functional screening. It is worth emphasizing that 2 μM warfarin was always included in the cell culture medium during all these procedures to prevent apoptosis of the reporter cells resulting from carboxylation of FIXgla-Fas due to the residual vitamin K in the cell culture medium.

For warfarin-resistant VKR screening, the lentiviral Brunello library transduced FIXgla-Fas/HEK293-Cas9 cells were harvested by trypsinization and filtered with 30 μm CellTrics filter (Sysmex America, Inc., Lincolnshire, IL) to obtain a well separated single-cell suspension. A total of $5 \times 10^7$ cells were frozen down as the control for genomic DNA extraction, the rest of the cells were seeded onto ten 245 mm × 245 mm dishes at a density of $4 \times 10^7$ cells per dish for functional screening. On the following day, 5 μM warfarin and 11 μM vitamin K were added to the cell culture medium as the final concentrations, and the cells were shaken on an orbital shaker at ~10 rpm to optimally separate the apoptotic cells and the attached survival cells. Medium were changed every 8 h with fresh medium containing 5 μM warfarin and 11 μM vitamin K to remove the apoptotic cells during functional screening. We optimized VKD apoptotic screening conditions empirically in an attempt to achieve maximal separation of the apoptotic cells and survival cells. Twenty-four hours post warfarin and vitamin K

treatments, the dishes were washed with PBS to remove loosely attached cells. Survival cells were harvest by trypsinization for genomic DNA extraction and next-generation sequencing (NGS) analysis. Two independent screens starting from lentiviral sgRNA library preparation, transduction followed by separated functional selections, genomic DNA extraction and sequencing were performed to account for stochastic noise. Normalized read counts for each sgRNAs (normalized as counts per million, CPM) were used to perform statistical analysis.

## Genomic DNA sequencing and candidate gene validation

Genomic DNA (gDNA) from the control and the functional screened cells were isolated using the Quick-DNA MidiPrep Plus kit (Zymo Research, Cat. No. D4075) according to the manufacturer's instructions. The concentration of gDNA was determined using NanoDrop UV spectrophotometer. The library of sgRNA target regions were amplified from gDNA by PCR[17]. All the gDNA harvested from the screen was used for PCR amplification. PCR products were pooled for each sample and purified using Zymo-Spin V columns with Reservoir (Zymo Research #C1016-25) according to the manufacturer's instructions. Purified PCR product was separated on a 2% agarose gel and the fragment of ~270 bp product was recovered by gel extraction using QIAquick Gel Extraction Kit (QIAGEN #28704) according to the manufacturer's directions. PCR products were quantified using NanoDrop for next-generation sequencing by Broad Institute Genomic Services. Data analysis was performed by The Bioinformatics and Analytics Research Collaborative (BARC) at the University of North Carolina at Chapel Hill. Cutadapt v.2.9 was used for trimming the original sequences from NGS results, Bowtie2 v.2.4.1 was used for sequence alignment, and Samtools v.1.13 was used for generating the counts of sgRNA using "idxstats" option.

For validation of the candidate genes, the individual lentivirial plasmid lentiCRISPR v2 containing the top two sgRNAs for each candidate gene, one sgRNA targeting GGCX (positive control), and one non-targeting sgRNA (negative control) were co-transfected with the helper plasmids in HEK293T cells as described above using Xfect transfection reagent. Lentivirus was prepared from the individual sgRNA construct and transduced into a different reporter cell line, FIXgla-*Met*.Luc/HEK293, to minimize the off-targeting effect. After puromycin selection for 7 days, survival cells were incubated with 11 μM vitamin K and the carboxylation of FIXgla-*Met*.Luc was evaluated with luciferase activity assay.

## Positioning of substrates into AlphaFold model

The model of FSP1 (AIFM2) was generated by AlphaFold (model number: AF-Q9BRQ8-F1-model_v2; https://alphafold.ebi.ac.uk/entry/Q9BRQ8)[65,66]. Positions of the NADH and FAD molecules were based on superposition of the structure of Ndi1 a type-II NADH dehydrogenases (NDH-2) from Saccharomyces cerevisiae (PDBcode 4G73) onto the FSP1 model (rmsd 2.1 Å over 318 Cα atoms)[41] in Coot[67]. For perspective, vitamin K was manually placed into the model of FSP1 based on the position of the quinone in Ndi1 in Coot. Coordinates 4G73 were chosen to position the NADH, FAD, and vitamin K due to the higher sequence identity in the crystal structure of Ndi1 to FSP1 (29%), resolution limit (2.5 Å) and these coordinates have both cofactors (NADH and FAD) and the substrate quinone bound. Cartoon figures of the superpositions were generated with PyMOL[68].

## Cell-based VKR activity assay of FSP1 and its mutants

Functional studies of FSP1 and its mutants were performed in FIXgla-PC/HEK293, DKO, and TKO reporter cells, as previously described[19]. The assay is based on the ability of the endogenous or exogenously expressed FSP1 or its mutants to convert vitamin K to $KH_2$ in HEK293 cells to support VKD carboxylation. Briefly, plasmid DNA of pcDNA3.1 containing the cDNA of wild-type or mutant FSP1 was transiently

expressed in the above reporter cells using Xfect transfection reagent. At four hours post-transfection, the transfection medium was replaced by a complete medium containing 11 μM vitamin K or a series concentrations of vitamin K (vitamin K titration) for activity assay. For warfarin-resistance study, the transfection medium was replaced by complete medium containing 11 μM vitamin K with increasing concentrations of warfarin. After incubation for 48 h, the cell culture medium was collected and directly used for ELISA to determine the level of carboxylated reporter protein, FIXgla-PC. The half-maximal effective concentration of vitamin K ($EC_{50}$) was determined using GraphPad software.

## Cell-based VKR enzyme kinetics study

Cell-based enzyme kinetics studies of HQNO inhibiting FSP1 was performed in DKO reporter cells. Briefly, DKO cells were seeded in a 96-well plate one day prior to the substrate/inhibitor treatment. On the following day, the cell culture medium was replaced with complete medium containing increasing concentrations of vitamin K with or without a certain concentration of HQNO as indicated in the figure. After incubation for 24 h, the cell culture medium was harvested and used directly to measure the levels of carboxylated FIXgla-PC. Experimental data was used to determine the inhibition model by Lineweaver–Burk plot using GraphPad software.

## VKR in vitro activity assay

Enzymatic activity of FSP1 converting K vitamins to their hydroquinone forms was evaluated directly by measuring different forms of vitamin K as the substrate using a conventional reversed-phase HPLC assay[69]. Cell pellets of HEK293 or these cells transiently expressing FSP1 or other NAD(P)H-dependent oxidoreductases were resuspended in TBS buffer (50 mM Tris-HCl, pH 7.6, 150 mM NaCl, 1x protease inhibitor cocktail) and lysed by sonication on ice. The reactions were performed in 200 μL ice-cold assay buffer containing 1 mg/mL cell lysate protein, 0.5% CHAPS, 2 mM NADH and 100 μM K vitamins at 30 °C for 1 h. When needed, 5 μM warfarin was included in the reaction mixture. The reaction was terminated by the addition of 500 μL of isopropanol. K vitamins were extracted with 500 μL n-hexane. After a brief centrifugation, the upper organic phase containing the vitamins was transferred to a 2-ml brown vial and dried with nitrogen. Then, a total of 500 μL of HPLC mobile phase was added to dissolve K vitamins and the sample was analyzed by HPLC[69]. To protect $KH_2$ from being oxidized back to vitamin K during the procedure, 2 mM DTT was included as the antioxidant during the extraction and following the steps.

## VKR activity assay using the activity-based fluorescence probes

The reduction of the activity-based fluorescent probes of vitamin K (VK-ASM) and $CoQ_{10}$ (NIR-ASM) by FSP1 were determined by in vitro activity assay. Fluorescent probe VK-ASM or NIR-ASM was dissolved in DMSO to obtain the 2 mM stock solution. FSP1 or its mutations were transiently overexpressed in TKO cells and cell lysis was prepared by sonication in TBS buffer. FSP1 reduction activity was performed in TBS buffer containing 0.5 mg/mL cell lysis protein, 0.5% CHAPS, 1 mM NADH and 20 μM VK-ASM or NIR-ASM in a 96 well black flat-bottom microplate at 30 °C. Fluorescence intensity (excitation wavelength: 488 nm and emission wavelength: 644 nm) was monitored every minute for 30 min with 5 s of shaking between intervals using Molecular Devices plate reader SpectraMax M5.

## Chemical synthesis of the activity-based fluorescence probe of vitamin K

**General experimental information.** Unless stated otherwise commercially available chemicals were used without further purification. Methanesulfonic acid, acetonitrile, piperidine, ethanol, pyridine and dichloromethane were purified and dried following literature procedures[70]. Malononitrile was purified following Ferris et al.[71] and 4-acetamidobenzaldehyde was recrystallized from water and dried under high vacuum (0.2 torr) overnight. Moisture sensitive reactions were carried out with oven (160 °C) dried glassware under an argon atmosphere. Argon was dried by passing through a drying tube with 3 Å molecular sieves and Direrite™. Thin layer chromatography (TLC) analysis was performed on Merck TLC silica gel 60 F254 plates, observed with ultraviolet light (254 nm), and/or staining with potassium permanganate, vanillin, or phosphomolybdic acid stains. Distilled solvents and silica gel (230-400 mesh) were used for flash column chromatography. NMR spectra were recorded using either a Bruker AS500 (500 MHz, 126 MHz), AV500 (500 MHz, 126 MHz), or a Bruker AV300 (300 MHz, 75 MHz) instrument (Supplementary Fig. 13). Chemical shifts (δ) are reported in parts per million (ppm), and internally referenced to solvent (CDCl₃ −7.26 ppm for ¹H NMR and 77.16 ppm for ¹³C NMR; DMSO-$d_6$ −2.49 ppm for ¹H NMR and 39.52 ppm for ¹³C NMR). Coupling constants (J) are given in hertz (Hz). Multiplicity was reported as follows: s = singlet, br s = broad singlet, d = doublet, t = triplet, q = quartet, quin = quintuplet, sext = sextuplet, spt = septet, m = multiplet. Low resolution electrospray ionization mass spectrometry measurements (LRESIMS) were measured on a Bruker HCT 3D Ion Trap spectrometer with a Bruker ESI source and recorded in positive ion mode. High resolution electrospray ionization (HRESIMS) accurate mass measurements were measured on a Bruker MicroOTOF-Q spectrometer with a Bruker ESI source and recorded in positive ion mode (Supplementary Fig. 14). Melting points were measured by Digimelt MPA161 SRS apparatus. Gas chromatography/mass spectroscopy (GC/MS) was measured by Shimadzu GCMS-QP5000 Benchtop Mass Spectrometer System.

**3-Methyl-3-(3-methyl-1,4-dioxo-1,4-dihydronaphthalen-2-yl)buta-noic acid (S4).** Following Gong et al.[45] Anhydrous hydroquinone **S1** (342 mg, 2 mmol) and 3,3-dimethylacrylic acid methyl ester **S2** (240 mg, 2.1 mmol) were added simultaneously to a hot solution (70 °C) of anhydrous methanesulfonic acid (2.4 mL) under an argon atmosphere. The reaction mixture was stirred at 70 °C for 2 h and then cooled to room temperature. The mixture was diluted with ice-cooled water (400 mL) and extracted with EtOAc (3 × 100 mL). The combined organic layers were washed with saturated NaHCO₃ solution (3 × 100 mL), water (3 × 100 mL), brine, and then dried over Na₂SO₄. After filtration, the solvent was removed in vacuo, and the residue was purified using an Isolera One system (SNAP Cartridge silica 20 g, elute: 10-30% EtOAc/Petroleum Ether) to afford crude **S3** as a brown solid (71 mg). ¹H NMR (500 MHz, CDCl₃) δ 8.24−8.20 (m, 1H), 8.04−7.99 (m, 1H), 7.55−7.48 (m, 2H), 2.68 (s, 2H), 2.54 (s, 3H), 1.55 (s, 6H) ppm. ¹³C NMR (126 MHz, CDCl₃) δ 168.4, 145.9, 140.7, 126.5, 126.4, 126.3, 123.8, 123.3, 122.0, 120.4, 115.5, 46.4, 36.0, 27.9, 15.0 ppm. HRMS (ESI): m/z calculated for: $[C_{16}H_{17}O_3H]^+$ (M + H)⁺: 257.1172; found: 257.1165.

To a stirring solution of crude **S3** (71 mg) in acetonitrile (14 mL) was added cerium ammonium nitrate (607 mg, 1.1 mmol) in one potion. The reaction was stirred at room temperature for 3 h and was then quenched by addition of H₂O (10 mL). The mixture was extracted with EtOAc (3 × 20 mL), and the combined organic layers were then washed with brine and dried over Na₂SO₄. After filtration, the solvent was removed in vacuo and the residue was dissolved in NaOH aqueous solution (1 M), and washed with DCM (3 × 30 mL). The aqueous layer was subsequently acidified by HCl (0.5 M) to pH = 2, and the suspension extracted with DCM (3 × 30 mL). The combined extracts were washed with brine, dried over Na₂SO₄ and concentrated in vacuo to obtain the title compound **S4** as a yellow semi-solid (19 mg, 3% yield over two steps). ¹H NMR (500 MHz, CDCl₃) δ 7.99−7.95 (m, 1H), 7.82−7.78 (m, 1H), 7.63−7.58 (m, 2H), 3.09 (s, 2H), 2.27 (s, 3H), 1.48 (s, 6H) ppm. ¹³C NMR (126 MHz, CDCl₃) δ 188.5, 185.6, 178.3, 154.7, 142.4, 134.5, 133.4, 132.8, 131.5, 125.9, 125.7, 47.6, 38.7, 29.1, 15.2 ppm. HRMS

(ESI): *m/z* calculated for: $[C_{16}H_{16}O_4Na]^+$ $(M + Na)^+$: 295.0941; found: 295.0932.

**2-(3,5,5-Trimethylcyclohex-2-en-1-ylidene)malononitrile (S7).** Following Liu et al.[72] To a solution of isophorone (691 mg, 5 mmol) and malononitrile (429 mg, 6.5 mmol) in anhydrous EtOH (10 mL) was added piperidine (60 μL, 0.6 mmol) and glacial acetic acid (34 μL, 0.6 mmol) under an argon atmosphere. After the mixture was refluxed at 90 °C overnight, it was then cooled to room temperature and concentrated in vacuo. The residue was dissolved in DCM (50 mL), washed with $H_2O$ (3 ×30 mL), dried over $Na_2SO_4$, and the organic solvent was removed in vacuo. The crude was purified by flash column chromatography on the silica gel (10% EtOAc/Petroleum Ether) to afford the title compound as a white solid (635 mg, 68% yield). m.p. 74.8–75.8 °C (Lit[73]. 72–74 °C). $^1$H NMR (300 MHz, $CDCl_3$) δ 6.62 (h, *J* = 1.4 Hz, 1H), 2.51 (s, 2H), 2.17 (dq, *J* = 2.0, 1.0 Hz, 2H), 2.03 (q, *J* = 1.0 Hz, 3H), 1.01 (s, 6H) ppm. $^{13}$C NMR (75 MHz, $CDCl_3$) δ 170.5, 159.9, 120.7, 113.3, 112.5, 78.4, 45.8, 42.8, 32.5, 28.0, 25.4 ppm. GC/MS EI m/z (%): 144 (100, $[MC_3H_6]^+$), 157 (56, $[MC_2H_5]^+$), 186 (35, $M^+$), 171 (27, $[MCH_3]^+$), 130 (13, $[MC_4H_8]^+$), 117 (13, $[MC_5H_9]^+$), 103 (6, $[MC_6H_{11}]^+$).

**(E)-N-(4-(2-(3-(Dicyanomethylene)−5,5-dimethylcyclohex-1-en-1-yl)vinyl)phenyl)acetamide (S9).** Following Liu et al.[74] To a solution of **S7** (186 mg, 1 mmol) and 4-acetamidobenzaldehyde (163 mg, 1 mmol) in anhydrous acetonitrile (4 mL) was added piperidine (40 μL, 0.05 mmol) under an argon atmosphere. The mixture was stirred at reflux for 2 h. On cooling to room temperature, the orange precipitation was collected and washed with ice-cooled acetonitrile (3 × 10 mL) to obtain the title compound as an orange solid (254 mg, 77% yield). m.p. > 260 °C. $^1$H NMR (300 MHz, DMSO-$d_6$) δ 10.12 (s, 1H), 7.67–7.59 (m, 4H), 7.36–7.16 (m, 2H), 6.83 (s, 1H), 2.60 (s, 2H), 2.53 (s, 2H), 2.06 (s, 3H), 1.01 (s, 7H) ppm. $^{13}$C NMR (75 MHz, DMSO-$d_6$) δ 170.3, 168.5, 156.2, 140.7, 137.5, 130.6, 128.7, 127.9, 122.1, 118.9, 114.0, 113.2, 75.6, 42.3, 38.2, 31.7, 27.4, 24.1 ppm. LRMS (ESI): m/z $(M + H)^+$: 332.

**(E)−2-(3-(4-Aminostyryl)−5,5-dimethylcyclohex-2-en-1-ylidene) malononitrile (S10).** Following Liu et al.[74] Compound **S9** (123 mg, 0.4 mmol) was dissolved in a mixture of EtOH and 10 M HCl (12 mL, v/v = 1:2) and refluxed at 105 °C for 13 h. On cooling to room temperature, the reaction was neutralized with saturated $NaHCO_3$ solution and extracted with EtOAc (3 × 40 mL). The combined organic phases were dried over $Na_2SO_4$ and concentrated in vacuo. The residue was passed through a short pad of silica gel (eluted by 100% DCM) to give crude title compound (101 mg) as a dark colored solid. The crude **S10** was used in the next step without further purification. LRMS (ESI): m/z $(M^+)$: 289.

**(E)-N-(4-(2-(3-(Dicyanomethylene)−5,5-dimethylcyclohex-1-en-1-yl)vinyl)phenyl)−3-methyl-3-(3-methyl-1,4-dioxo-1,4-dihydronaphthalen-2-yl)butanamide (S11).** To a solution of compound **S4** (27 mg, 0.1 mmol) in anhydrous DCM (1 mL) was added EDC·HCl (29 mg, 0.15 mmol), followed by pyridine (100 μL), at room temperature under an argon atmosphere. After stirring for 20 min, **S10** (29 mg, 0.1 mmol) was added, and the mixture was stirred at rt for 14 h. The reaction was then diluted with $H_2O$ (20 mL) and extracted with EtOAc (3 ×15 mL). The combined organic layers were washed with brine, dried over $Na_2HSO_4$ and concentrated in vacuo. The residue was purified by flash column chromatography on the silica gel (5:30:65 EtOAc/DCM/Petroleum Ether) to obtain the title compound as a yellow colored solid (12 mg, 22%). Additional, purification was carried out for biological evaluation i.e., **S11** (9 mg) was dissolved in DCM (200 μL) and then *n*-hexane (2 mL) was added. On mixing a yellow precipitate appeared which was collected by filtration and washed with a 1% DCM/*n*-hexane solution. The yellow solid was then dried in vacuo for 8 h. m.p. 121 °C (Dec.). $^1$H NMR (500 MHz, $CDCl_3$) δ 8.03–7.98 (m, 1H), 7.86–7.81 (m, 1H), 7.66–7.60 (m, 2H), 7.43–7.36 (m, 4H), 6.96 (d, *J* = 16.0 Hz, 1H), 6.86 (d, *J* = 16.0 Hz, 1H), 6.80 (s, 1H), 3.16 (s, 2H), 2.58 (s, 2H), 2.43 (s, 2H), 2.33 (s, 3H), 1.59 (s, 6H), 1.06 (s, 6H) ppm. $^{13}$C NMR (126 MHz, $CDCl_3$) δ 189.3, 185.6, 170.3, 169.4, 155.2, 154.1, 142.3, 139.2, 136.5, 134.6, 133.5, 132.9, 131.7, 131.6, 128.5, 128.3, 126.1, 125.8, 123.4, 119.9, 113.7, 113.0, 78.5, 51.0, 43.1, 39.5, 39.3, 32.2, 29.6, 28.2, 15.2 ppm. HRMS (ESI): *m/z* calculated for: $[C_{35}H_{34}N_3O_3]^+$ $(M + H)^+$: 544.2595; found: 544.2599; $[C_{35}H_{33}N_3O_3Na]^+$ $(M + Na)^+$: 566.2414; found: 566.2421.

**Reporting summary**
Further information on research design is available in the Nature Portfolio Reporting Summary linked to this article.

## Data availability
Alphafold model of FSP1 can be found at https://alphafold.ebi.ac.uk/entry/Q9BRQ8. Coordinates for Ndi1 can be found at https://www.rcsb.org/structure/4G73. Coordinates of the superposition of the substrates from coordinates 4G73 into the Alphafold model are available in the supplementary material. Tissue specificity for the FSP1 gene is available from The Human Protein Atlas Database at https://www.proteinatlas.org/ENSG00000042286-AIFM2/tissue. All other data associated with this study can be found in the paper, the Supplementary materials or the Source Data file. Source Data are included in this paper. Source data are provided with this paper.

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

## Acknowledgements

We thank Dr. Lee Pedersen for helpful discussions, Dr. Alain Laederach for initial analysis of the next generation sequence results, Dr. Corbin Jones, Dr. Hemant Kelkar, and Dr. Zhaohui Man from the Bioinformatics and Analytics Research Collaborative (BARC) at the University of North Carolina at Chapel Hill for next generation sequence data analysis, and Dr. Chuan-An Zhang for helping on the preparation of the amplified Brunello library preparation. This work was supported in part by the National Institutes of Health HL131690 to J.K.T. and D.W.S. and the Division of Intramural Research of the National Institute of Environmental Health Sciences, National Institutes of Health Grant 1ZIA ES102645 to L.C.P.; Y.L. and C.M.W. gratefully acknowledge financial support from the University of Queensland. The content is solely the responsibility of the authors and does not necessarily represent the official views of the National Institutes of Health.

## Author contributions

D.Y.J. created the reporter cell lines, performed library screening, made all mutation constructs, and performed cell-based activity studies. X.C. characterized the VKD apoptotic reporter cells and performed the in vitro activity assays. Y.L. and C.M.W. designed, synthesized and characterized the fluorescent probe. L.C.P. modeled FSP1 structure and its binding with cofactors and substrate, suggested mutations for functional study. D.W.S. and J.K.T. conceived the study. J.K.T. coordinated and designed the study, analyzed and interpreted the data, and wrote the manuscript. All authors read and agreed on the content of the paper.

## Competing interests

The authors declare no competing interests.
