## [Peer Review File · Nature Communications]

A genome-wide CRISPR-Cas9 knockout screen identifies FSP1 as the warfarin-resistant vitamin K reductaseREVIEWER COMMENTS

Reviewer #1 (Remarks to the Author):

Using a genome-wide CRISPR/Cas screen the authors identified ferroptosis suppressor protein-1 (FSP1) as the missing warfarin-resistant vitamin K reductase, corroborating recent findings published by Mishima and colleagues earlier this year. The authors generated an elegant cell-based screening model where they fused the vitamin K-dependent Gla domain of the coagulation factor IX to the extracellular N-terminal region of the death receptor Fas. Using this system, the authors found besides the major downstream signaling partner of Fas signaling, FSP1 as the enzyme conferring warfarin resistance. While knockout or pharmacological inhibition of FSP1 had a strong impact on ferroptosis sensitivity and vitamin K recycling, inhibiting the ubiquinone reductase dihydroorotate dehydrogenase (DHODH) failed to contribute to vitamin K dependent carboxylation. Overall, this is an interesting study clearly supporting recent results by Mishima et showing that FSP1 is the long-sought warfarin-resistant vitamin k quinone reductase. Although the conclusions are convincing, there are several points that need to be addressed. My specific comments are:

- 1) Cells expressing the chimeric receptor need to be maintained in the presence of warfarin. Would pan-caspase inhibitors also prevent the dying of cells induced by the receptor?
- 2) Since the study was performed using just one cultured cell line (HEK), the contribution of FSP1 to the VK cycle in cells other than HEK remains unclear. As such, the role of FSP1 in other cells derived from the VK-target organs, such as the liver and bone, should be examined.
- 3) One of the main limitations of the study is the lack of in vivo data, although the authors provide a number of cell- and enzyme-based data concerning the role of FSP1 as a potent vitamin K reductase. As such, studies on tissue samples or primary cells are required to support the physiological contribution of FSP1 in VK reduction. In particular, evaluating the individual in vivo contribution of VKOR and FSP1 in VK quinone reduction would be key.
- 4) The authors propose that ER-resident FSP1 reduces VK. However, the cells expressing the N-myristoylation defective G2A mutant still present VKD carboxylation activity (Fig 4b). Since N-myristoylation of FSP1 is known to be necessary for membrane binding, the results suggest that the ability of FSP1 binding to membranes is dispensable for VK reduction in cells. Thus, the authors need to examine whether G2A mutant still localizes to the ER membrane to provide direct proof whether or not FSP1-mediated VK reduction indeed occurs at the ER membrane. Provided that FSP1 may reduce VK at a subcellular site other than the ER, how can FSP1 get access to VK quinone from ER-localized VKOR complex, and how can VK hydroquinone be passed to GGFX in the ER?
- 5) Although both FSP1 and VKOR function as a VK reductase, the efficiency and capacity for VK reduction seem to vary greatly between FSP1 and VKOR (as illustrated in Fig. 3b). What factors (substrate recognition, cellular localization, etc) may account for the different affinity of both enzymes required for VKD carboxylation?
- 6) As mentioned, triple KO cells (VKOR^{-/-}, VKORL1^{-/-} and FSP1^{-/-}) still show some residual minor VKR activity. The authors should examine whether the remaining VKR activity is actually attributed to NQO1, for instance by ablating NQO1 and/or using NQO1 inhibitors.
- 7) The authors used HQNO as an FSP1 inhibitor, although there are no reports that HQNO is a direct FSP1 inhibitor. Ref #31 does not contain relevant data. Either the authors provide the appropriate reference or include data showing that HQNO indeed inhibits FSP1 in a cell-free system.
- 8) Several immunoblotting data validating the KO and forced expression of FSP1 are missing: double/triple KO (in Fig 3), stable re-expression of FSP1 (Fig 1f) and forced expression of each FSP1 mutant (in Fig 4).
- 9) On page 14, the authors raise the issue that targeting the FSP1 pathway may induce bleeding risks; however, FSP1 KO mice do not show any bleeding abnormalities under normal housing condition as shown by Mishima et al (Nature 2022). As such, FSP1 inhibition alone without warfarin co-treatment does not seem to induce bleeding risks.
- 10) Although the authors claim that FSP1 is expressed widely among the organs (Ext Fig 5), a previous report showed it is actually predominantly expressed in adipose tissues (Mol Cell 2020, PMID: 31952989).

Minor:

- Fig 2a should read blasticidin.
- Due to the low resolution of the images in Fig 4a, it is hard to discern the colocalization of FSP1 with the respective organelle.
- On page 15 the authors state that reduced VK also functions as a mitochondrial electron carrier; however, the function of VK is not observed in mammals (Sci Rep 2019, PMID: 31024065).
- On page 8, line 14, the authors raised two possibilities for the cause of increased EC50 for VK carboxylation in VKOR KO cells: i) lower efficiency for vitamin K reduction by FSP1, or ii) due to impaired ability to recycle VK epoxide by VKOR – the authors should provide some insights which theory is indeed more plausible.
- It would be appropriate to refer to Mishima et al already in the abstract as this is the first report identifying FSP1 as the warfarin-resistant VK reductase.

Reviewer #2 (Remarks to the Author):

This contribution by Jin et al reported the identification of ferroptosis suppressor protein 1 (FSP1) as warfarin-resistant vitamin K reductase (VKR) utilizing a genome-wide CRISPR-Cas9 knockout screening with a newly developed vitamin K-dependent apoptotic reporter cell line (FIXgla-Fas/HEK293). A new activity-based fluorescent probe of vitamin K, namely VK-ASM, was designed and synthesized for investigating the enzymatic activity of FSP1 to substrates. Overall, this is a very strong, thorough piece of work. These findings provide new findings and insights into selectively controlling the biological processes mediated by vitamin K.

Some small issues recommended being addressed are listed below.

On page 5, the authors described that they use 11 μ M vitamin K and 5 μ M warfarin in the cell-based assays for screening of the warfarin-resistant VKR. Why the authors used the mentioned concentration for vitamin K and warfarin? Does the curve in Fig 1d help to set the concentration for vitamin K? From Fig 1d, we could see that the curve of the cell viability has reached the lower plateau at 1 μ M; thus I wonder why 11 μ M of vitamin K was used in the cell-based screening assay. I suggest the addition of 1-2 sentences (or re-writing existing sentences) to clarify this for the reader.

In Fig 4f and extended data Fig.9a, I suggest the word “NADH” around the arrow would be better to be described as “NAD(P)H”, because NQO1 uses either NADH or NADPH as the cofactor for its bio-reduction process in cells.

Point-by-point response to reviewers' comments

Reviewer #1

Using a genome-wide CRISPR/Cas screen the authors identified ferroptosis suppressor protein-1 (FSP1) as the missing warfarin-resistant vitamin K reductase, corroborating recent findings published by Mishima and colleagues earlier this year. The authors generated an elegant cell-based screening model where they fused the vitamin K-dependent Gla domain of the coagulation factor IX to the extracellular N-terminal region of the death receptor Fas. Using this system, the authors found besides the major downstream signaling partner of Fas signaling, FSP1 as the enzyme conferring warfarin resistance. While knockout or pharmacological inhibition of FSP1 had a strong impact on ferroptosis sensitivity and vitamin K recycling, inhibiting the ubiquinone reductase dihydroorotate dehydrogenase (DHODH) failed to contribute to vitamin K dependent carboxylation. Overall, this is an interesting study clearly supporting recent results by Mishima et showing that FSP1 is the long-sought warfarin-resistant vitamin k quinone reductase. Although the conclusions are convincing, there are several points that need to be addressed.

Response: We are grateful for the reviewer's positive evaluation of our manuscript and appreciation for developing a cell-based screening model with convincing conclusions that identify FSP1 as the warfarin-resistant vitamin K reductase. During the preparation of our manuscript, we became aware of the article published by Mishima *et al.* (*Nature*, 2022, **608**:778-83, online published on August 3rd, 2022), who used different methods to reach the same conclusion.

My specific comments are:

1) Cells expressing the chimeric receptor need to be maintained in the presence of warfarin. Would pan-caspase inhibitors also prevent the dying of cells induced by the receptor?

Response: Our reporter cells expressing the chimeric receptor protein (FIXgla-Fas) are sensitive to vitamin K-dependent (VKD) carboxylation (Figure 1). Even the residual amount of vitamin K in the cell culture medium is sufficient to stimulate apoptosis of the reporter cells.

Therefore, we included warfarin in the cell culture medium to prevent VKD apoptosis during the maintenance of the reporter cells. We also showed that VKD apoptosis of the reporter cells was mediated by the caspase-dependent apoptotic pathway (Figure 1 and Supplementary Table 1). Therefore, once VKD apoptosis of the reporter cell is activated, functional interruption of any proteins in the caspase-dependent pathway, such as inhibition of caspases by the pan-caspase inhibitors (as the reviewer mentioned), would stop apoptosis and prevent the death of the reporter cells. However, as long as VKD apoptosis of the reporter cells is not activated, the pan-caspase inhibitors do not affect the cell growth. Therefore, it is not necessary to include the pan-inhibitors for the maintenance of the reporter cells. Importantly, pan-caspase inhibitors interfere with the specific designed VKD apoptosis that create false-positive cells as background for the functional screening.

2) Since the study was performed using just one cultured cell line (HEK), the contribution of FSP1 to the VK cycle in cells other than HEK remains unclear. As such, the role of FSP1 in other cells derived from the VK-target organs, such as the liver and bone, should be examined.

Response: Our goal of this study was to identify the so-called antidotal enzyme that rescued patients from warfarin poisoning by vitamin K administration. Although this enzyme's function was discovered in the 1940s, its identity was recalcitrant to identification. We performed a genome-wide CRISPR-Cas9 knockout screening in human embryonic kidney 293 (HEK293) cells, a mammalian cell line widely used for studying VKD carboxylation and coagulation factors production (*Methods Mol. Biol.*, 2018, **1674**:49-61). We previously demonstrated that HEK293 cells have significant antidotal enzyme activity (*Blood*, 2011, **117**:2967-74), which is critical for the genome-wide screening of the target enzyme. We identified FSP1 as the unknown antidotal enzyme (warfarin-resistant VKR). We further confirmed the identity of VKR by 1) characterization of FSP1 using several gene-knockout cell lines of HEK293 cells, 2) conventional *in vitro* activity assays, and 3) an activity assay using our novel activity-based fluorescence probe. Significantly, while preparing our manuscript, the identity of VKR was published by Mishima *et al.* using a totally different approach (*Nature*, 2022, **608**:778-83). In fact, Mishima *et al.* have demonstrated the ability

of FSP1 to reduce vitamin K for ferroptosis protection and VKD carboxylation in several other human and mouse samples including: human cancer cell lines, human liver cell line (HepG2), mouse fibroblasts, and mouse tissues. Thus, it seems already clear that FSP1 is indeed the canonical vitamin K reductase in human cells other than HEK293.

3) One of the main limitations of the study is the lack of in vivo data, although the authors provide a number of cell- and enzyme-based data concerning the role of FSP1 as a potent vitamin K reductase. As such, studies on tissue samples or primary cells are required to support the physiological contribution of FSP1 in VK reduction. In particular, evaluating the individual in vivo contribution of VKOR and FSP1 in VK quinone reduction would be key.

Response: The majority of the data in this study, from genome-wide screening to functional characterization of FSP1 and its mutants for vitamin K reduction (Fig. 1d, Fig. 2c, d, f, Fig. 3a to 3f, Fig. 4b, e, Fig. 5b to 5e, and Sup Fig. 4), were obtained from HEK293 cells. This human cell line is widely used for studying VKD carboxylation and coagulation factors production (*Methods Enzymol.*, 2017, **584**:349-94; *Methods Mol. Biol.*, 2018, **1674**:49-61). In this study, VKR activity of FSP1 and its mutants were evaluated by measuring carboxylation efficiency of a native protein substrate, the Gla domain of coagulation factor IX, using the endogenous carboxylation machinery of HEK293 in live cells (*Blood*, 2011, **117**:2967-74). Therefore, results reported in this study are very different from studies describing VKR activity as assessed by conventional *in vitro* activity assays under artificial conditions using purified enzyme or crude extract from tissue samples or cultured cells (*Methods Enzymol.*, 1990, **186**:287-301, *Biochem. J.*, 2013, **456**:47-54). For example, conventional *in vitro* VKR activity was determined by following the decrease of absorbance at 340 nm of the cofactor NADH or by directly measuring the production of vitamin K hydroquinone from crude extract in the presence of detergents (*Methods Enzymol.*, 1990, **186**:287-301, *Biochem. J.*, 2013, **456**:47-54). In this context, we believe that our data can be categorized as *in vivo* data, as the reviewer suggested (*from tissue samples or primary cells*). Additionally, *the physiological contribution of FSP1 in vitamin K reduction from tissue samples or primary cells* (as the reviewer suggested) are now available from the recent study by Mishima *et al.* (*Nature*, 2022, **608**:778-83, Fig. 4 and Extended Data Fig. 10).

This study aimed to identify the enzyme that had defied purification efforts by our group and others for decades. Although Mishima *et al.* published the identity of VKR in *Nature* about two weeks before our submission, we used a different method to go after the enzyme itself and arrived at the same conclusion. Indeed, there may be years to fully understand the role of FSP1 in the vitamin K cycle. At present, everything points to FSP1 functioning as a salvage vitamin K reductase that is resistant to warfarin inhibition. It seems that “*studies on tissue samples or primary cells to support the physiological contribution of FSP1 in VK reduction*” (as the reviewer suggested) are not necessary for the following reasons: 1) we are looking for a particular warfarin-resistant enzymatic activity for vitamin K reduction, which has been clearly demonstrated in human cells (HEK293); 2) *in vivo* evidence for FSP1 reducing vitamin K in (mouse) tissues and several human cell lines are available from the recent study by Mishima *et al.* (*Nature*, 2022, **608**:778-83). Additionally, we believe that “*evaluating the individual in vivo contribution of VKOR and FSP1 in VK quinone reduction*”, as suggested by the reviewer, is beyond the scope of identifying the warfarin-resistant VKR for this study.

4) *The authors propose that ER-resident FSP1 reduces VK. However, the cells expressing the N-myristoylation defective G2A mutant still present VKD carboxylation activity (Fig 4b). Since N-myristoylation of FSP1 is known to be necessary for membrane binding, the results suggest that the ability of FSP1 binding to membranes is dispensable for VK reduction in cells. Thus, the authors need to examine whether G2A mutant still localizes to the ER membrane to provide direct proof whether or not FSP1-mediated VK reduction indeed occurs at the ER membrane. Provided that FSP1 may reduce VK at a subcellular site other than the ER, how can FSP1 get access to VK quinone from ER-localized VKOR complex, and how can VK hydroquinone be passed to GGCX in the ER?*

Response: Based on two recent studies (*Nature*, 2019, **575**:688-92 and 693-98), N-myristoylation of FSP1 is essential for the anti-ferroptotic function of FSP1, presumably by recruiting it to the plasma membrane. Additionally, these two studies show that mutating the myristoylation site of FSP1 (G2A) abolishes its anti-ferroptotic activity. It is worth noting that

in one of the studies, Doll *et al.* also demonstrated that the subcellular distribution of FSP1 overlaps with an endoplasmic reticulum (ER) marker (Figure 2e in *Nature*, 2019, **575**:693-98), which is consistent with our observations (Figure 4a),

We showed that the G2A mutant is fully active for VKD carboxylation (Figure 4b) in the ER lumen. This strongly suggests that disrupting the myristoylation of FSP1 does not affect its quinone reductase activity or its ER localization. Per the reviewer's suggestion, we assessed the subcellular distribution of the G2A mutant in HEK293 cells. As shown in the following figure, unlike wild-type FSP1 (Figure 4a), the G2A mutant is distributed throughout the cell. Nevertheless, it appears that the G2A mutant overlaps with the ER marker, which is consistent with the observation of *Doll et al.* (Figure 2e in *Nature*, 2019, **575**:693-98).

Additionally, Nguyen *et al.* has demonstrated that myristoylation of FSP1 is essential for its lipid droplet association but not mitochondrial localization (*Mol Cell*, 2020, **77**:600-17, Figure 2D). Taken together, these results suggest that disrupting the myristoylation of FSP1 does not prevent its ER localization. Therefore, the VKD carboxylation activity of the G2A mutant does not involve transferring vitamin K hydroquinone from other organelles to the ER.

It is worth noting that the subcellular localizations of FSP1 described in the literature are not consistent. For example, Doll *et al.* (*Nature*, 2019, **575**:693-98) and this study show that FSP1 is distributed in multiple perinuclear membrane compartments including the ER, while results from Bersuker *et al.* (*Nature*, 2019, **575**:688-92) demonstrated that FSP1 does not co-localize with the ER marker. Whether this disagreement is due to these studies using different cells, and whether the myristoylation of FSP1 determines its multiple subcellular localizations or there are multiple factors that affect the subcellular localization of FSP1, are interesting and important questions for future studies.

5) Although both FSP1 and VKOR function as a VK reductase, the efficiency and capacity for VK reduction seem to vary greatly between FSP1 and VKOR (as illustrated in Fig. 3b). What factors (substrate recognition, cellular localization, etc) may account for the different affinity of both enzymes required for VKD carboxylation?

Response: We agree with the reviewer that clarifying the significant differences of the efficiency and capacity for vitamin K reduction between VKOR and FSP1 is a critical question. As we discussed in the revised manuscript (page 17, lines 350-353), VKOR reduces vitamin K by the active site free cysteines (Cys132 and Cys135), which are oxidized to a disulfide after vitamin K reduction. The disulfide in the active site needs to be reduced to free cysteines by a yet unknown physiological reductant to reactivate VKOR. But importantly, FSP1 does not have an “active site residue” that is directly involved in vitamin K reduction. Instead, it shuttles reducing equivalents from NADH to FAD where reduced FAD reduces vitamin K. Without knowing the physiological reductant of VKOR and other possible cofactors for FSP1, it is difficult to clarify the efficiency and capacity differences between these two enzymes for vitamin K reduction.

6) As mentioned, triple KO cells (*VKOR*^{-/-}, *VKORL1*^{-/-} and *FSP1*^{-/-}) still show some residual minor VKR activity. The authors should examine whether the remaining VKR activity is actually attributed to NQO1, for instance by ablating NQO1 and/or using NQO1 inhibitors.

Response: We thank the reviewer for suggesting further clarification of the residual VKR activity that we observed in the triple knockout cells. As the reviewer suggested, the best approach for this could be knocking out the potential candidate genes, such as NQO1, in the triple knockout cells. However, as we stated in the manuscript that “the alternative vitamin K reduction pathways, including VKOR and possibly NQO1, only contribute negligibly to VKD carboxylation in our test conditions” (Supplementary Fig. 4b) (page 10, lines 206-207). Given that the goal of this study is to identify the unknown warfarin-resistant VKR, and that we and Mishima *et al.* (*Nature*, 2022, **608**:778-83) have now shown that FSP1 is the key player for warfarin-resistant vitamin K reduction, we think the suggested experiment by the reviewer for clarifying the minor contribution of the residual VKR activity (Supplementary Fig. 4b) is beyond the scope of this study.

7) The authors used HQNO as an FSP1 inhibitor, although there are no reports that HQNO is a direct FSP1 inhibitor. Ref #31 does not contain relevant data. Either the authors provide the appropriate reference or include data showing that HQNO indeed inhibits FSP1 in a cell-free system.

Response: We thank the reviewer for catching our citing error. Direct evidence of HQNO inhibiting FSP1 was reported by Elguindy *et al.*, (*JBC*, 2015, **290**:20815-26, reference 23 of the original manuscript). In this study, FSP1 was referred to as AMID (AIFM2) and was categorized as a type 2 NADH:ubiquinone oxidoreductase (NDH-2). Elguindy *et al.* reported that HQNO is a strong inhibitor of FSP1 with an IC₅₀ of ~1 μM (*JBC*, 2015, **290**:20815-26). For clarification, we now cite this reference in FSP1 inhibition study in the revised manuscript (page 14, second paragraph and page 15. Line 306).

8) Several immunoblotting data validating the KO and forced expression of FSP1 are missing: double/triple KO (in Fig 3), stable re-expression of FSP1 (Fig 1f) and forced expression of each FSP1 mutant (in Fig 4).

Response: The validation of double-gene knockout reporter cells in Figure 3 is available in our previous study (*JTH*, 2013, 11:1556-64) and the reference is cited in the revised manuscript (page 8, line 156). Immunoblotting for knocking out FSP1 in HEK293 cells (including triple KO) can be found in Fig. 2e, as we used the same sgRNA knocking out FSP1 in the same cell with the same approach for screening. The immunoblotting data of stable re-expression of FSP1 (Fig. 2f, not Fig. 1f) and forced expression of each FSP1 mutant (Fig. 4) were included (as shown below) and discussed in the revised manuscript (page 12, second paragraph, and page 13. First paragraph).

Fig. 2f

Fig. 4b and 4e

9) On page 14, the authors raise the issue that targeting the FSP1 pathway may induce bleeding risks; however, FSP1 KO mice do not show any bleeding abnormalities under normal housing condition as shown by Mishima *et al* (Nature 2022). As such, FSP1 inhibition alone without warfarin co-treatment does not seem to induce bleeding risks.

Response: The related statement has been modified and further discussed in the revised manuscript (page 16, second paragraph).

10) Although the authors claim that FSP1 is expressed widely among the organs (Ext Fig 5), a previous report showed it is actually predominantly expressed in adipose tissues (Mol Cell 2020, PMID: 31952989).

Response: We believe that our statement on FSP1's tissue distribution has no conflict with the result from the recent study Nguyen *et al.* (Mol Cell, 2020, 77:600-17, PMID: 31952989). We stated in the manuscript that "FSP1 appears to have a low tissue specificity (Supplementary Fig. 5)..." The tissue specificity of human FSP1/AIFM2 in Supplementary Figure 5 was obtained from **The Human Protein Atlas Database**. Results in this Figure 5

shows that FSP1 in adipose tissue is ~4-fold higher than in most of other tissues, which is consistent with the results from Nguyen *et al.* (*Mol Cell*, 2020, **77**:600-17, PMID: 31952989). However, the overall specificity of tissue distribution of FSP1 is low, as concluded from the database in the Supplementary Figure 5.

Minor:

- Fig 2a should read *blasticidin*.

Response: This has been corrected.

- Due to the low resolution of the images in Fig 4a, it is hard to discern the colocalization of FSP1 with the respective organelle.

Response: We have revised the confocal image presentation in the revised manuscript to make the co-localization of FSP1 clearer (as shown below).

- On page 15 the authors state that reduced VK also functions as a mitochondrial electron carrier; however, the function of VK is not observed in mammals (*Sci Rep* 2019, PMID: 31024065).

Response: As we cited in our statement, the function of vitamin K as a mitochondrial electron carrier was reported by Vos *et al.* (*Science*, 2012, **336**:1306-10, reference 35 in the original manuscript). However, as the reviewer pointed out, a later study from Cerqua *et al.* suggests that “Vitamin K2 cannot substitute Coenzyme Q10 as electron carrier in the mitochondrial respiratory chain of mammalian cells” (*Sci Rep* 2019, PMID: 31024065). The discrepancy between these two studies could be due to the different models that were used in these studies, as pointed out by Cerqua *et al.* (*Sci Rep* 2019, PMID: 31024065). Nevertheless, it does not affect our citation of the work from Vos *et al.* (*Science*, 2012, **336**:1306-10) who demonstrated that vitamin K could serve as an electron carrier in mitochondria.

- On page 8, line 14, the authors raised two possibilities for the cause of increased EC50 for VK carboxylation in VKOR KO cells: i) lower efficiency for vitamin K reduction by FSP1, or ii) due to impaired ability to recycle VK epoxide by VKOR – the authors should provide some insights which theory is indeed more plausible.

Response: In the revised manuscript, we have explained that the lower efficiency of VKD carboxylation in DKO cells was due to the cells losing the ability to recycle K_{epo} by VKOR, which supports the essential role of VKOR in VKD carboxylation as previously reported (*JBC*, 2005, **280**:31603-7; *Blood*, 2005, **106**:3811-5; *Biochemistry*. 2006, **45**:5587-98) (page 9, lines 169-172).

- It would be appropriate to refer to Mishima *et al* already in the abstract as this is the first report identifying FSP1 as the warfarin-resistant VK reductase.

Response: The discovery of FSP1 as warfarin-resistant VKR by Mishima *et al.* is included in the abstract per the reviewer’s suggestion.

Reviewer #2

This contribution by Jin et al reported the identification of ferroptosis suppressor protein 1 (FSP1) as warfarin-resistant vitamin K reductase (VKR) utilizing a genome-wide CRISPR-Cas9 knockout screening with a newly developed vitamin K-dependent apoptotic reporter cell line (FIXgla-Fas/HEK293). A new activity-based fluorescent probe of vitamin K, namely VK-ASM, was designed and synthesized for investigating the enzymatic activity of FSP1 to substrates. Overall, this is a very strong, thorough piece of work. These findings provide new findings and insights into selectively controlling the biological processes mediated by vitamin K.

Response: We are gratified for the reviewer's positive review of our manuscript.

Some small issues recommended being addressed are listed below. On page 5, the authors described that they use 11 uM vitamin K and 5 uM warfarin in the cell-based assays for screening of the warfarin-resistant VKR. Why the authors used the mentioned concentration for vitamin K and warfarin? Does the curve in Fig 1d help to set the concentration for vitamin K? From Fig 1d, we could see that the curve of the cell viability has reached the lower plateau at 1 uM; thus I wonder why 11 uM of vitamin K was used in the cell-based screening assay. I suggest the addition of 1-2 sentences (or re-writing existing sentences) to clarify this for the reader.

Response: Per the reviewer's suggestion, we have now included the rationale for including a higher concentration of vitamin K in the functional screening (page 6, second paragraph).

In Fig 4f and Supplementary Fig.9a, I suggest the word "NADH" around the arrow would be better to be described as "NAD(P)H", because NQO1 uses either NADH or NADPH as the cofactor for its bioreduction process in cells.

Response: We thank the reviewer for making the suggestion, and as such, NADH was replaced by NAD(P)H in these two figures.

REVIEWER COMMENTS

Reviewer #1 (Remarks to the Author):

The authors provide some additional cell-based data and have properly answered many of the reviewer's comments. Yet, several issues remain and need to be addressed. In particular, as the authors refer to multiple results reported by other groups, it would be at least appropriate to tone down the novelty of their findings and also consider to rephrase the title.

1. As the present study examined the role of FSP1 as VKR only in one single type of cells, it is generally not sufficient to conclude that FSP1 is the VKR in the broader, let alone in vivo, context. As the authors repeatedly state, the requested evidence largely relies on the data reported by other groups. Thus, the authors need to consider this limitation of the present manuscript. For instance, it would be more than appropriate to change the phrase "reveals" to "validate" in the title or the like.

2. Concerning Fig. 3e-f, the reviewer still deems that determining the contribution of NQO1 in residual VKR activity in triple KO cells is important and not beyond the scope of the present study. As commented previously, the authors should examine whether the remaining VKR activity can be indeed attributed to NQO1, for instance by genetically ablating NQO1 and/or using NQO1 inhibitors (at least). I hope the authors agree but this is certainly not an elaborate experiment.

3. Validation of loss of FSP1 expression in the triple KO (TKO) cells is still missing, although the FSP1 on single FSP1 KO cells was shown by WB and genomic sequencing (Fig. 2f and Supplementary Fig. 3). Since the triple KO cells were newly generated in the present study, the data confirming the loss of FSP1 expression in TKO cells is crucial. Even if the same sgRNAs were used, the parental cells are different between single KO and TKO cells, with all the known caveats of clonality and off-target effects.

4. The fluorescence images were revised but they are hardly convincing as such. For instance, in Fig. 4a, the signal of the mitochondrial marker looks aggregated and does not show the usual mitochondrial pattern. Consider to redo the confocal microscopy figures.

5. In the additional data of supplemental Fig. 8, the expression of the His-fused FSP1 D41A mutant was not examined by WB.

6. Consider to include the fluorescence image of G2A FSP1-expressing cells (as shown in the response letter) in the manuscript.

Reviewer #2 (Remarks to the Author):

The authors have made important efforts to properly address the questions raised by the referees. I have no additional comments and recommend it for publication in Nature Communications.

Point-by-point response to reviewers' comments-V2

Reviewer #1

The authors provide some additional cell-based data and have properly answered many of the reviewer's comments. Yet, several issues remain and need to be addressed. In particular, as the authors refer to multiple results reported by other groups, it would be at least appropriate to tone down the novelty of their findings and also consider to rephrase the title.

1. As the present study examined the role of FSP1 as VKR only in one single type of cells, it is generally not sufficient to conclude that FSP1 is the VKR in the broader, let alone in vivo, context. As the authors repeatedly state, the requested evidence largely relies on the data reported by other groups. Thus, the authors need to consider this limitation of the present manuscript. For instance, it would be more than appropriate to change the phrase "reveals" to "validate" in the title or the like.

Response: We thank the reviewer for their suggestion to change the manuscript title. Although the reviewer suggested to change the word "reveals" to "validate or the like", we feel "identifies" is more accurate as this study was done independently and unknowingly of the Mishima *et al.*'s study and was not an attempt to validate their results. This study did identify independently the role of FSP1 as the warfarin-resistant vitamin K reductase. Therefore, we think the word "identifies" is more accurate for the following reasons.

a). Publication timeline. Although the requested evidence required by the reviewer to support FSP1 as warfarin-resistant vitamin K reductase are available from the previous publication by Mishima *et al.*, our experiment design, data collection, and manuscript preparation were completed before Mishima *et al.*'s paper was published. The Mishima *et al.* paper was published on August 3rd, 2022, and the first submission of this manuscript was on August 12, 2022.

b). Experiment design and data presentation. As discussed in the manuscript, VKD carboxylation is a nonlethal post-translational modification with no apparent phenotypic consequences in cell growth, it is therefore impossible for genome-wide screening to assist in identification of unknown enzymes in the vitamin K cycle. We spent several years

optimizing our reporter cell line to adapt the gene-wide screening, which included expressing different chimeric Gla reporter-proteins on the cell surface and using FACS (Fluorescence-Activated Cell Sorting) and antibody-based magnetic separation for genome-wide loss-of-function screening. Finally, we were very fortunate to identify a unique reporter-protein FIXgla-Fas with VKD apoptotic screening. We then used genome-wide CRISPR-Cas9 knockout screening and associated functional studies to confirm that FSP1 is the warfarin-resistant VKR. As FSP1 was reported as a ferroptosis suppressor mediated by reducing CoQ10 (references 32 and 33), we compared the effect of two CoQ10-mediated ferroptosis pathways on VKD carboxylation. All these experimental designs and data collection were completed before knowing that FSP1 was the enzyme to reduce vitamin K.

c). Research community. After the Pre-print of this manuscript was posted on Research Square (<https://www.researchsquare.com/article/rs-2039668/v1>), Dr. Mishima (the first author of the Nature paper to show FSP1 was VKR) tweeted on his tweeter saying that “Tie and Stafford’s group reported FSP1 is the vitamin K reductase by CRISPR screening. They *independently* came to the same conclusion as us in a different way” (see the screenshot below).

[REDACTED]

2. Concerning Fig. 3e-f, the reviewer still deems that determining the contribution of NQO1 in residual VKR activity in triple KO cells is important and not beyond the scope of the present study. As commented previously, the authors should examine whether the remaining VKR activity can be indeed attributed to NQO1, for instance by genetically ablating NQO1 and/or using NQO1 inhibitors (at least). I hope the authors agree but this is certainly not an elaborate experiment.

Response: Based on the following reasons, we believe it is unnecessary to clarify the contribution of NQO1 to the residual VKR activity in the triple KO (TKO) cells in this study. a) the residual VKR activity in the triple KO cells is trivial (Supplementary Figure 4c, as shown below); b) the goal of this study is to identify the unknown warfarin-resistant VKR, c) FSP1 is the warfarin-resistant VKR and the major contributor for vitamin K reduction (Fig. 3 and results from Mishima *et al.*, *Nature*, 2022, **608**:778-83).

3. Validation of loss of FSP1 expression in the triple KO (TKO) cells is still missing, although the FSP1 on single FSP1 KO cells was shown by WB and genomic sequencing (Fig. 2f and Supplementary Fig. 3). Since the triple KO cells were newly generated in the present study, the data confirming the loss of FSP1 expression in TKO cells is crucial. Even if the same sgRNAs were used, the parental cells are different between single KO and TKO cells, with all the known caveats of clonality and off-target effects.

Response: As per the reviewer's suggestions, western blot results showing knockout of FSP1 in TKO cells are now included in the Supplementary Figure 4a.

4. The fluorescence images were revised but they are hardly convincing as such. For instance, in Fig. 4a, the signal of the mitochondrial marker looks aggregated and does not show the usual mitochondrial pattern. Consider to redo the confocal microscopy figures.

Response: FSP1 was initially identified as a mitochondria-associated protein, named as AIFM2 (Apoptosis Inducing Factor Mitochondria-associated 2). Results from this study suggest that FSP1 is a vitamin K reductase that can support VKD carboxylation i.e., a post-translational modification occurs in the endoplasmic reticulum (ER). Fig. 4a is provided to clarify the ER subcellular location of FSP1 using mitochondria (the well-accepted organelle where FSP1 is located) marker as a control. Although the mitochondria marker in Fig. 4a is not ideal, the figure nevertheless clearly demonstrates the overlap location of FSP1 (green image) with mitochondria (red image).

5. In the additional data of supplemental Fig. 8, the expression of the His-fused FSP1 D41A mutant was not examined by WB.

Response: The western blot result for the expression of the D41A mutant was already presented in Fig. 4b.

6. Consider to include the fluorescence image of G2A FSP1-expressing cells (as shown in the response letter) in the manuscript.

Response: We thank the reviewer for their suggestion to include the fluorescence image of G2A in the manuscript. However, we are unable to find a good fit of the G2A subcellular location image in this manuscript, mainly because the focus of this study concentrated on the identification of FSP1 as the vitamin K reductase that supports VKD carboxylation in the ER. While myristoylation at residue G2 was shown to be essential for FSP1's plasma

membrane localization and anti-ferroptotic activity, which are not closely related to the focus of the study.

Reviewer #2

The authors have made important efforts to properly address the questions raised by the referees. I have no additional comments and recommend it for publication in Nature Communications.

Response: We gratefully acknowledge the reviewer's positive and supportive comments.